# DISENTANGLED INTERLEAVING VARIATIONAL ENCODING

## ABSTRACT

Conflicting objectives present a considerable challenge in interleaving multi-task learning, necessitating the need for meticulous design and balance to ensure effective learning of a representative latent data space across all tasks without mutual negative impact. Drawing inspiration from the concept of marginal and conditional probability distributions in probability theory, we design a principled and well-founded approach to disentangle the original input into marginal and conditional probability distributions in the latent space of a variational autoencoder. Our proposed model, Deep Disentangled Interleaving Variational Encoding (Deep-DIVE) learns disentangled features from the original input to form clusters in the embedding space and unifies these features via the cross-attention mechanism in the fusion stage. We theoretically prove that combining the objectives for reconstruction and forecasting fully captures the lower bound and mathematically derive a loss function for disentanglement using Naïve Bayes. Under the assumption that the prior is a mixture of log-concave distributions, we also establish that the Kullback-Leibler divergence $D_{KL}$ between the prior and the posterior is upper bounded by a function minimized by the minimizer of the cross entropy loss, informing our adoption of radial basis functions (RBF) and cross entropy with interleaving training for DeepDIVE to provide a justified basis for convergence. Experiments on two public datasets show that DeepDIVE disentangles the original input and yields forecast accuracies better than the original VAE and comparable to existing state-of-the-art baselines.

## 1 INTRODUCTION

In multi-objective deep learning, gradients from different objectives can conflict, when the different loss terms induce competing gradient directions during training of the network. Balancing these gradients to ensure stable and effective learning is a significant challenge prompting the development of methods to mitigate this issue, such as Liu et al. (2021); Yu et al. (2020); Sener & Koltun (2018) which solve an additional optmization problem before each gradient update step, to manipulate conflicting gradients before the update. In contrast to such methods which usually involve some additional computations to search for non-conflicting gradient updates, we derive the overall loss function from the log likelihood of a pre-existing data point, hence it follows that these objectives exhibit no mutual conflict.

**Contribution.** In this paper, our primary contribution lies in extending the original VAE architecture to include the use of labelled data for a forecasting use case. We motivate the use of semi-supervised learning considering the fact that while it is common to employ a sequence as input to a time series forecasting model, other available labels that classify the entire sequence are often overlooked. For this task, the benefits of our proposed Deep Disentangled Interleaving Variational Encoding (DeepDIVE) architecture are two-fold. (i) Building upon the statistical foundations provided by the Variational Autoencoder (VAE) (Kingma & Welling, 2014) for learning deep latent-variable models and corresponding inference models for reconstruction, DeepDIVE also provides a principled and well-founded semi-supervised framework for learning deep latent-variable models and corresponding inference models which is useful for reconstruction, forecasting, and classification. (ii) We also believe that probabilistic latent variable models such as VAEs hold great potential for facilitating autonomous planning and resource allocation in situations characterized by uncertainty, mirroring real-world scenarios.

To address this task, DeepDIVE learns a disentangled representation space without conflicting objectives. The main contributions of our study are summarized as follows:

- To obtain objectives that are not mutually conflicting, we theoretically extend the VAE architecture originally intended for generative reconstruction tasks to a generative forecasting case by proving that combining the objectives for reconstruction and forecasting fully captures the lower bound.

- Drawing inspiration from the concept of marginal and conditional probability distributions in probability theory, we mathematically derive a loss function to disentangle the model input in the latent space by applying Bayes' theorem with the "naïve" assumption of independence among the marginal dimensions conditional on the original input. This marginalisation approach decomposes the input into its more manageable constituent elements, yielding univariate distributions that can be useful for human-interpretable analysis of data representations while maintaining the absence of conflict across the different objectives.

- Following these derivations, we design DeepDIVE to learn disentangled features from the original input, and further propose to unify these features via the cross-attention mechanism in the fusion stage. The disentangled embedding space easily lends itself to exploration of how important features of the data is encapsulated and encoded.

**Paper Organization.** We structure our paper as follows: In Section 2 we introduce notation related to the VAE loss function and review related works for disentangling the latent space. In Section 3 we derive the loss function for extending the VAE to accommodate a forecasting scenario, and establish that the minimizer of the cross entropy loss also minimizes the $D_{KL}$ under certain assumptions. Following this, in Section 4 we propose DeepDIVE and empirically validate its benefits in Section 5. Finally, we summarize our main contributions and provide further details and more complete proofs in the Appendix.

## 2 RELATED WORKS

### 2.1 VARIATIONAL AUTOENCODER AND THE REPARAMETERIZATION TRICK

Our work is built upon the VAE (Kingma & Welling, 2014), originally intended for reconstruction tasks. In their paper, Kingma & Welling (2014) assume that each arbitrary data sample $x^{(i)}$ is generated from some random process, where the likelihood $p_{\theta^*}(x^{(i)}|z^{(i)})$ involves a hidden random variable $z^{(i)}$ generated from some prior $p_{\theta^*}(z)$, and $p_{\theta^*}(x|z)$ and $p_{\theta^*}(z)$ come from parametric families of distributions $p_\theta(x|z)$ and $p_\theta(z)$ with the true parameters $\theta^* \in \Theta$ unknown. In the following we omit index $i$ for convenience. To learn an approximation of the (intractable) true posterior $p_{\theta^*}(z|x)$, the authors proposed the VAE architecture, which we can view as a combination of 3 components:

- Probabilistic decoder $p_\theta(x|z)$ parameterized by learned generative model parameters $\theta \in \Theta$ approximates the likelihood and maps latent random variable $z$ to a conditional distribution over the data $x$.

- Prior $p_\theta(z)$ over the latent random variables $z$, where any learnable parameters are also considered to be part of $\theta$.

- Probabilistic encoder $q_\phi(z|x)$ parameterized by learned recognition model parameters $\phi \in \Phi$ approximates the intractable true posterior, ie. $q_\phi(z|x) \approx p_\theta(z|x)$, and maps data observations $x$ to a distribution over the possible values of $z$ in the latent space.

Observing that the log likelihood of the model can be expressed as the following sum:

$$\log p_\theta(x) = \mathcal{L}(\theta, \phi; x) + D_{KL}(q_\phi(z|x) \parallel p_\theta(z|x)), \tag{1}$$

where the second RHS term representing the $D_{KL}$ between the approximate and true posterior is always non-negative, we have that $\log p_\theta(x)$ is lower bounded by the variational lower bound $\mathcal{L}(\theta, \phi; x)$:

$$\mathcal{L}(\theta, \phi; x) = \mathbb{E}_{q_\phi(z|x)}[\log p_\theta(x|z)] - D_{KL}(q_\phi(z|x) \parallel p_\theta(z)) \tag{2}$$

Thus, since direct maximization of the log likelihood equation 1 is not possible due to the intractable term $p_\theta(z|x) = p_\theta(x|z)p_\theta(z)/p_\theta(x)$ where $p_\theta(x) = \int_Z p_\theta(x|z)p_\theta(z)\,dz$, we instead jointly train the generative model parameters $\theta$ and recognition model parameters $\phi$ to maximize the lower bound.

The reparameterization trick was proposed as a solution to the high variance of the expected reconstruction error (first RHS term) in equation 2. By introducing a new random variable $\epsilon$, the originally random $z$ can thus be formulated as a deterministic function of $\epsilon$, instead of directly sampling $z$ from $Z \sim q_\phi(z|x)$ as in the usual Monte Carlo estimation.

Attempts made on disentangled VAEs have mostly focused on weighing or further decomposition of the $D_{KL}$.

## 2.2 DISENTANGLED VAES

### 2.2.1 WEIGHTED $D_{KL}$

Viewing the latent space as an embedding space and the outputs of the encoder as embedded data, the latent space of a VAE can be made interpretable when restricted to low dimensions, or by projecting a multi-dimensional latent space into a lower dimension via PCA. The prevalent methods to disentangle the VAE encourage the model to learn these independent factorizations directly by simply increasing the influence of the regularization term (second RHS term) in equation 2. Higgins et al. (2017) proposed the $\beta$-VAE by constraining the divergence between the prior and approximate posterior to be less than some $\epsilon > 0$, which the authors reformulated as a Lagrangian under KKT conditions. Since the divergence between the distributions is at most $\epsilon$, and the prior is chosen with 0 covariance, thus the covariance between dimensions in the latent space is also close to 0. The Lagrangian dual problem:

$$\min_\beta \max_{\theta,\phi} \mathbb{E}_{q_\phi(z|x)}[\log p_\theta(x|z)] - \beta * D_{KL}(q_\phi(z|x) \,\|\, p_\theta(z)) + \beta * \epsilon \tag{3}$$

$$\text{st.}\beta \geq 0 \tag{4}$$

has $\beta * \epsilon \geq 0$ by complimentary slackness. Since the optimal value of $\beta$ depends on the hyperparameter $\epsilon$, hence $\epsilon$ is removed from the equation and $\beta$ is treated as a hyperparameter. Thus, the model is trained by maximizing the lower bound $\mathbb{E}_{q_\phi(z|x)}[\log p_\theta(x|z)] - \beta * D_{KL}(q_\phi(z|x) \,\|\, p_\theta(z))$ of the Lagrangian formulation, where the authors set hyperparameter $\beta > 1$ to drive the divergence to be closer to 0.

Burgess et al. (2018) and Sankarapandian & Kulis (2021) extend this idea to learn factors in order of importance by encouraging the values of the regularization term to increase as training progresses. The former's Bottleneck VAE pressures the divergence to be close to a controllable value $C$ that is gradually increased from zero, while the latter's $\beta$-annealed VAE proposed a gradual decrease of $\beta$, from $\beta$-VAE's loss formulation. Bae et al. (2023) proposed a separate extension with the Multi-Rate VAE, which constructs the rate-distortion curve to learn the optimal parameters corresponding to various values of $\beta$ in a single training run.

### 2.2.2 FURTHER DECOMPOSITION OF $D_{KL}$

Various sets of decompositions of the evidence lower bound $\mathcal{L}(\theta, \phi; x)$ have been proposed in Hoffman & Johnson (2016), of which the third (Average term-by-term reconstruction minus KL to prior) holds the most interest, as it further dissects the divergence term:

$$\frac{1}{n} \sum_{i=1}^n D_{KL}(q_\phi(z|x^{(i)}) \,\|\, p_\theta(z)) = D_{KL}(q_\phi(z) \,\|\, p_\theta(z)) + (\log N - \mathbb{E}_{Z \sim q_\phi}[H(N|Z = z)]) \tag{5}$$

where the authors used $p_\theta(z^{(i)}) = p_\theta(z) \,\forall\, i \in \{1, ..., n\}$ as the embeddings for all data samples assume the same prior. The first RHS term is termed the marginal KL to prior, while the second RHS term is the mutual information between the index, or rather the specific data sample, and its corresponding embedding, also known as its code.

Extending from this derivation, Chen et al. (2018) uses a straightforward equation to further dissect the marginal KL to prior from Hoffman & Johnson (2016) via

$$D_{KL}(q_\phi(z) \parallel p_\theta(z)) = \mathbb{E}_{Z \sim q_\phi} \left[ \log \frac{q_\phi(z)}{\prod_{j=1}^{d} q_\phi(z_j)} \frac{\prod_{j=1}^{d} q_\phi(z_j)}{p_\theta(z)} \right] \tag{6}$$

to obtain the total correlation and dimension-wise KL, and proposed the $\beta$-TCVAE based on the claim that the total correlation term is the most important term in this derivation. $d$ in equation 6 refer to the number of latent dimensions. The authors also proposed the Mutual Information Gap (MIG) for measuring disentanglement.

## 3 LOSS FUNCTION DERIVATIONS

We use similar notations as in section 2.1, where we also consider the case for a single data sample and omit index $i$ to lighten notation. For the remainder of this paper, consider the normalized lookback window $x = X_{t-L+1:t} \in \mathbb{R}^L$ in section 2.1, and the normalized forecast window $y = X_{t+1:t+H} \in \mathbb{R}^H$. Let $a = [\, a_1, \ldots, a_{n_1} \,] \in \mathbb{R}^{n_1}$ and $b = [\, b_1, \ldots, b_{n_2} \,] \in \mathbb{R}^{n_2}$ form the latent space, ie. $z = [\, a \parallel b \,] = [\, a_1, \ldots, a_{n_1}, b_1, \ldots, b_{n_2} \,] \in \mathbb{R}^{n_1+n_2}$. Without loss of generality, we fix $q_\phi(a,b|x) = q_\phi(a|b,x)q_\phi(b|x)$, and thus also refer to $a$ and $b$ as the conditional and marginal dimensions respectively, since the marginal probability distribution in each dimension in $b$ is only conditional on the input data point, while the conditional probability distribution in each dimension in $a$ is conditional on both $b$ and the original input.

**Proposition 1.** *Given jointly continuous random variables $x$ and $y$, joint probability density function $p(x,y) = p(y|x)p(x)$, the log likelihood of the joint distribution can be written as*

$$\log p_\theta(x,y) =: \mathcal{L}(\theta, \phi; x, y) + D_{KL}(q_\phi(a,b|x) \parallel p_\theta(a,b|x,y)) \tag{7}$$

*where the Evidence Lower Bound can be written as*

$$\mathcal{L}(\theta, \phi; x, y) \tag{8}$$

$$= \mathbb{E}_{A,B \sim q_\phi}[\log p_\theta(y|a,b,x)] + \mathbb{E}_{A,B \sim q_\phi}[\log p_\theta(x|a,b)] + \mathbb{E}_{A,B \sim q_\phi}\left[\log \frac{p_\theta(a,b)}{q_\phi(a,b|x)}\right] \tag{9}$$

$$=: \textit{forecast loss} + \textit{reconstruction loss} - D_{KL}(q_\phi(a,b|x) \parallel p_\theta(a,b)) \tag{10}$$

*Similar to equation 2, $\mathcal{L}(\theta, \phi; x, y)$ in equation 9 is also a lower bound on the log-likelihood in equation 7.*

*Sketch of Proof:* Proposition 1 extends almost directly from the derivation of the original loss function for the VAE shown in Kingma & Welling (2014). Notably, the log-likelihood of the model $\log p_\theta(x,y)$ now includes both historical data $x$ and future data $y$, as we aim to maximize the likelihood of the joint density of the time series.

The full proof of proposition 1 is shown in Appendix A.2.1.

Proposition 1 shows that the log-likelihood of the joint distribution $p_\theta(x,y)$ can be written as the sum of the evidence lower bound $\mathcal{L}(\theta, \phi; x, y)$ and the divergence $D_{KL}(q_\phi(a,b|x) \parallel p_\theta(a,b|x,y))$ between the approximate and true posterior of the latent variables of the generative model. It follows that the VAE, originally formulated for reconstruction, can also be extended to a forecasting case.

**Assumption 1.** *As with the usual case in a VAE, we make the assumption that we choose a prior such that the dimensions in the latent space are independent.*

*Remark:* A consequence of this assumption is that $a$ and $b$ in the prior are independently distributed, ie. $p_\theta(a,b) = p_\theta(a)p_\theta(b)$.

**Proposition 2.** *Then, under assumption 1 the $D_{KL}$ between the prior and the approximate posterior can be further decomposed into the following marginal and conditional counterparts:*

$$D_{KL}(q_\phi(a,b|x) \parallel p_\theta(a,b)) = \mathbb{E}_{B \sim q_\phi}[D_{KL}(q_\phi(a|b,x) \parallel p_\theta(a))] + D_{KL}(q_\phi(b|x) \parallel p_\theta(b)) \tag{11}$$

*Sketch of Proof:* Applying chain rule on $q_\phi(a,b|x)$ and assumption 1 on $p_\theta(a,b)$ we get

$$D_{KL}(q_\phi(a,b|x) \parallel p_\theta(a,b)) = \int_A \int_B q_\phi(a|b,x)q_\phi(b|x) \log \frac{q_\phi(a|b,x)}{p_\theta(a)} \frac{q_\phi(b|x)}{p_\theta(b)} \, db \, da \tag{12}$$

Then, by the product rule of logarithms and linearity of integration, we have

$$\int_A \int_B q_\phi(a|b,x) q_\phi(b|x) \log \frac{q_\phi(a|b,x)}{p_\theta(a)} \frac{q_\phi(b|x)}{p_\theta(b)} \, db \, da \qquad (13)$$

$$= \int_B q_\phi(b|x) \int_A q_\phi(a|b,x) \log \frac{q_\phi(a|b,x)}{p_\theta(a)} \, da \, db + \int_B q_\phi(b|x) \log \frac{q_\phi(b|x)}{p_\theta(b)} \int_A q_\phi(a|b,x) \, da \, db \qquad (14)$$

from which we can integrate out $a$ in the second RHS term to get

$$\int_B q_\phi(b|x) \int_A q_\phi(a|b,x) \log \frac{q_\phi(a|b,x)}{p_\theta(a)} \, da \, db + \int_B q_\phi(b|x) \log \frac{q_\phi(b|x)}{p_\theta(b)} \int_A q_\phi(a|b,x) \, da \, db \quad (15)$$

$$= \mathbb{E}_{B \sim q_\phi}[D_{KL}(q_\phi(a|b,x) \parallel p_\theta(a))] + D_{KL}(q_\phi(b|x) \parallel p_\theta(b)) \qquad (16)$$

by definition of expectation.

The full proof of proposition 2 is shown in Appendix A.2.2.

**Assumption 2.** *Further, similar to Naïve Bayes, here we also make the "naïve" assumption of independence among the marginal dimensions conditional on the original input, ie.*

$$q_\phi(b_i, b_j|x) = q_\phi(b_i|x) q_\phi(b_j|x) \, \forall \, i, j \in \{1, ..., n_2\}, \, i \neq j \qquad (17)$$

*Remark:* We note that the independence assumption in Assumption 2 generally does not hold true in real-world situations, although this simplifying assumption often works well in practice.

*Further remark and intuition:* In the context of the derivation, this naïve conditional independence assumption is used to split the total marginal Kullback-Leibler divergence $D_{KL}(q_\phi(\prod_{i=1}^{n_2} b_i|x) \parallel p_\theta(\prod_{i=1}^{n_2} b_i))$ into the sum of divergences for individual marginal dimensions $\sum_{i=1}^{n_2} D_{KL}(q_\phi(b_i|x) \parallel p_\theta(b_i))$, each of which shares a common minimizer with the cross entropy for that dimension. Thus, it would be more precise to say that the sum of divergences for individual marginal dimensions only approximates the total marginal Kullback-Leibler divergence, and this approximation is exact when Assumption 2 holds. We believe that in the case when Assumption 2 does not hold, ie. some pairs are positively correlated while others are negatively correlated, the summation term may have an aggregating effect even if the estimates of the individual marginal divergences are inaccurate.

**Proposition 3.** *Given marginal dimensions $b_i$ and $b_j$ where $i, j \in \{1, ..., n_2\}$ and $i \neq j$, the $D_{KL}$ in the second RHS term of proposition 2 can be further decomposed to*

$$D_{KL}(q_\phi(b_i, b_j|x) \parallel p_\theta(b_i, b_j)) = D_{KL}(q_\phi(b_i|x) \parallel p_\theta(b_i)) + D_{KL}(q_\phi(b_j|x) \parallel p_\theta(b_j)) \qquad (18)$$

*under assumptions 1 and 2.*

*Sketch of Proof:* Proposition 3 follows from a direct application of the assumption 1 and assumption 2 to $p_\theta(b_i, b_j)$ and $q_\phi(b_i, b_j|x)$ respectively, along with the definition of the $D_{KL}$ to obtain

$$D_{KL}(q_\phi(b_i, b_j|x) \parallel p_\theta(b_i, b_j)) = \int_{B_i} \int_{B_j} q_\phi(b_i, b_j|x) \log \frac{q_\phi(b_i|x) q_\phi(b_j|x)}{p_\theta(b_i) p_\theta(b_j)} \, db_j \, db_i \qquad (19)$$

Applying the product rule of logarithms and linearity of integration before marginalizing out $b_i$ and $b_j$ on the resultant terms yields the desired result.

The full proof of proposition 3 is shown in Appendix A.2.3.

## 3.1 RELATION TO CROSS ENTROPY LOSS

Similar to the above sections, here we also consider the case for a single data sample and follow the same notation as the above sections. In this section, we will focus on the decoder end of the network, to establish a relationship between the $D_{KL}$ and the cross entropy loss. This rationalizes substituting the divergence term in the loss function with cross entropy loss when categorical labels are made available.

We first define the radial basis function (RBF) $\psi_k$ for the $k^{\text{th}}$ univariate RBF unit parameterized by centroid $\nu_k$ and scale $\tau_k$. For example, given an arbitrary variable $b$, the $k^{\text{th}}$ univariate Gaussian RBF

is defined by equation 20. $\nu \in \mathbb{R}^K$ and $\tau \in \mathbb{R}^K$ are learnable parameters that are also considered part of $\theta$.

$$\psi_k(b) = \frac{1}{\sqrt{2\pi\tau_k^2}} \exp\left\{-\frac{1}{2}\left(\frac{b - \nu_k}{\tau_k}\right)^2\right\} \tag{20}$$

Without loss of generality, we consider marginal dimension $b_i$ with $K_i$ classes, $i \in \{1, ..., n_2\}$, and denote them as $b$ and $K$ for ease of notation.

**Proposition 4.** *For dimension $b$ with $K$ classes, the $D_{KL}$ between the learned probabilistic encoder $q_\phi(b|x)$ and the prior $p_\theta(b)$ can be written as:*

$$D_{KL}(q_\phi(b|x) \parallel p_\theta(b)) = -H(B|X = x) - \int_B q_\phi(b|x)\log\sum_{k=1}^K p_\theta(b, k)\, db \tag{21}$$

*where $-H(B|X = x)$ is known as the conditional differential entropy, and the second RHS term*

$$-\int_B q_\phi(b|x)\log\sum_{k=1}^K p_\theta(b, k)\, db \leq \mathbb{E}_{B \sim q_\phi}\left[\sum_{k=1}^K Q(k)\left[-\log\frac{p_\theta(b, k)}{Q(k)}\right]\right] \tag{22}$$

*holds for any $Q(k)$ st. $0 < Q(k) \leq 1 \ \forall \ k \in \{1, ..., K\}$, $\sum_{k=1}^K Q(k) = 1$*

*Sketch of Proof:* The proof for proposition 4 first introduces $p_\theta(b, k) = Q(k)\frac{p_\theta(b,k)}{Q(k)}$ then uses Jensen's inequality for convex functions. Observe that the inequality is tight if $Q(k) = p_\theta(k|b)$.

The full proof of proposition 4 is shown in Appendix A.2.4.

Thus proposition 4 shows that the $D_{KL}$ of the marginal dimension between the learned probabilistic encoder and the prior $D_{KL}(q_\phi(b|x) \parallel p_\theta(b))$ is upper bounded by the sum of the conditional differential entropy and the expectation term (RHS of equation 22). From an information theoretic perspective, maximizing the entropy of the encoder output increases the information gain. Since this entropy term is constant with respect to the decoder, it follows that we can minimize the divergence by minimizing the expectation term in the upper bound.

**Assumption 3.** *For our prior $p_\theta(b, k) = p_\theta(b|k)p_\theta(k)$ we assume a finite mixture model with $K$ components, where each component has a simple parametric form (for example a Gaussian distribution), modelled by*

$$p_\theta(b|k) = \frac{1}{\sqrt{2\pi\tau_k^2}} \exp\left\{-\frac{1}{2}\left(\frac{b - \nu_k}{\tau_k}\right)^2\right\} \tag{23}$$

$$= \psi_k(b) \tag{24}$$

Denote $n$ to be the total number of data samples and $n_k$ to be the number of samples belonging to class $k$, $k \in \{1, ..., K\}$.

**Assumption 4.** *Under the assumption that the distribution of our training samples is representative of the true distribution of the entire dataset, the prior probability of component $k$ can be approximated by $p_\theta(k) \approx \frac{n_k}{n}$.*

By assumption 4, the prior $p_\theta(k)$ is constant with respect to $\nu$ and $\tau$.

**Proposition 5.** *(Necessity) Then, under assumption 3, the value of $\theta$ which minimizes the upper bound in proposition 4 satisfies*

$$\mathbb{E}_{B \sim q_\phi}\left[-\sum_{k=1}^K Q(k)\left(\frac{p_\theta(k)}{p_\theta(b, k)}\frac{\partial}{\partial\nu}\psi_k(b)\right)\right] = 0 \tag{25}$$

$$\mathbb{E}_{B \sim q_\phi}\left[-\sum_{k=1}^K Q(k)\left(\frac{p_\theta(k)}{p_\theta(b, k)}\frac{\partial}{\partial\tau}\psi_k(b)\right)\right] = 0 \tag{26}$$

*Sketch of Proof:* By the product rule of logarithms and linearity of expectation, we have

$$\mathbb{E}_{B \sim q_\phi}\left[\sum_{k=1}^{K} Q(k)\left[-\log \frac{p_\theta(b, k)}{Q(k)}\right]\right] = \mathbb{E}_{B \sim q_\phi}\left[-\sum_{k=1}^{K} Q(k) \log p_\theta(b, k) + \sum_{k=1}^{K} Q(k) \log Q(k)\right]$$

(27)

The proof for proposition 5 uses the fact that the global minimum on a function differentiable everywhere implies that the derivative is 0. Since functions parameterized by $\phi$ are constant with respect to $\theta$, differentiating the above expression with respect to $\theta$ we have

$$\mathbb{E}_{B \sim q_\phi}\left[-\sum_{k=1}^{K} Q(k)\frac{\partial}{\partial \nu} \log p_\theta(b, k)\right] = 0 \quad \text{and} \quad \mathbb{E}_{B \sim q_\phi}\left[-\sum_{k=1}^{K} Q(k)\frac{\partial}{\partial \tau} \log p_\theta(b, k)\right] = 0 \quad (28)$$

Applying chain rule and assumption 3 to the LHS terms of the above equations yields the desired result.

The full proof of proposition 5 is shown in Appendix A.2.5.

**Corollary.** *(Sufficiency) If $p_\theta(b|k)$ is selected such that:*

- *$p_\theta(b|k)$ is a valid probability distribution*

- *$p_\theta(b|k)$ is log-concave in its parameters*

*then the necessary conditions equation 25 and equation 26 become sufficient conditions for optimality, since this implies that the stationary point of the upper bound is a global minimum point.*

The proof of the sufficiency corollary is shown in Appendix A.2.6.

Let $j$ be the true class of $x$.

**Definition 1.** *We first define the cross entropy loss*

$$\mathcal{L}_{CE}(\phi, \theta; j) = \sum_{k=1}^{K} -\mathbf{1}_{k=j} \log p_\theta(k|b)$$
$$= -\log p_\theta(j|b)$$

Here we prove the case for 2 classes, where by definition of the sigmoid function we have $p_\theta(j|b) = \sigma(f(\boldsymbol{\psi}(b))) = \frac{1}{e^{-f(\psi(b))}+1}$. Observe also that the softmax function in the 2 class case $\frac{e^{y_0}}{e^{y_0}+e^{y_1}}$ reduces to the sigmoid if $y_0$ is fixed at 0.

**Proposition 6.** *Let $\Omega \subset \mathbb{R}$, $\psi_k(b) : \mathbb{R} \to \Omega \,\forall\, k \in \{1, ..., K\}$ and $f(x) : \Omega \to \mathbb{R}$ such that $\frac{\partial}{\partial x}f(x) \neq 0$. Then if there exists a point $\theta^*$ that minimizes the cross entropy loss for each class $k \in 1, \ldots, K$ respectively, this point must satisfy*

$$\frac{\partial}{\partial \nu}\boldsymbol{\psi}(b) = \mathbf{0} \tag{29}$$

$$\frac{\partial}{\partial \tau}\boldsymbol{\psi}(b) = \mathbf{0} \tag{30}$$

*Sketch of Proof:* The proof for proposition 6 follows a similar procedure to that for proposition 5. Observing that $\sigma(x) \neq 0$ and $1 - \sigma(x) \neq 0 \,\forall\, x \in \mathbb{R}$ simplifies the necessary conditions of the stationary points to equation 29 and equation 30.

The full proof of proposition 6 is shown in Appendix A.2.7.

**Corollary.** *The minimizer of the cross entropy loss also minimizes the upper bound (RHS of equation 22) of the $D_{KL}$ of the marginal dimension.*

*Proof:* By equation 29 and equation 30, $\theta^*$ also satisfies equation 25 and equation 26.

A graphical illustration of the role played by some of the key terms and assumptions in the derivations is included in Appendix A.2.8.

## 3.2 Relation to Mean Squared Error

The relation between the maximum likelihood estimator and mean squared error is well-known. Readers may refer to Murphy (2012) for an in-depth understanding.

## 3.3 Overall Loss Function

In this subsection, we combine the results from the previous propositions and corollaries to present the overall loss function.

**Theorem 1.** *Consider*

$$Q(k) = p_{\theta_t}(k|b)$$

*Then by proposition 1, proposition 2 and proposition 3 the negative of the Evidence Lower Bound can be written as*

$$-\mathcal{L}(\theta, \phi; x, y) \tag{31}$$
$$= -\mathbb{E}_{A,B \sim q_\phi}[\log p_\theta(y|a, b, x)] - \mathbb{E}_{A,B \sim q_\phi}[\log p_\theta(x|a, b)] \tag{32}$$
$$+ \mathbb{E}_{B \sim q_\phi}[D_{KL}(q_\phi(a|b, x) \parallel p_\theta(a))] + \sum_{i=1}^{n_2} D_{KL}(q_\phi(b_i|x) \parallel p_\theta(b_i)) \tag{33}$$

*Further, by proposition 4, proposition 3.1 and proposition 6 the fourth RHS term can be minimized by minimizing $\mathcal{L}_{CE}(\phi, \theta; j_i)$ for each individual label $i \in \{1, ..., n_2\}$. The first and second RHS terms can be minimized by minimizing $\mathcal{L}_{MSE}(\phi, \theta; y)$ and $\mathcal{L}_{MSE}(\phi, \theta; x)$ respectively. Assuming a Gaussian prior on the conditional dimensions, the third RHS term can be integrated analytically as shown in Appendix B of Kingma & Welling (2014).*

Theorem 1 dissects the lower bound of the log likelihood of the data point into four terms, each to be minimized. In the context of this multi-objective optimization problem, given that the constituent terms of the objective arise from the log likelihood of a pre-existing data point, it follows that these objectives exhibit no mutual conflict, and thus training the model on the loss function derived in theorem 1 should ensure effective learning of a representative latent data space across all tasks without mutual negative impact.

## 4 Deep Disentangled Interleaving Variational Encoding (DeepDIVE)

The derived loss function in theorem 1 plays a pivotal role in informing the architecture and training paradigm of DeepDIVE, specifically: (i) Employing cross entropy loss in place of $D_{KL}$ for the marginal dimensions, (ii) Utilizing interleaving training to optimize model performance, since cross entropy loss is only a bound on the $D_{KL}$, and (iii) Integrating a Gaussian radial basis function layer into the model, which satisfies the assumptions made in the sufficiency corollary.

Our latent space consists of $n_1 + n_2$ latent dimensions, where $n_2$ dimensions represent the marginal embedding features and are inputs to classifiers, and the remaining $n_1$ dimensions represent the other conditional embedding features. The classification from the latent space is employed as an auxiliary task in order to encourage clustering in the relevant dimension in the latent space, from which we can obtain the marginal probability distribution in each dimension conditional on the input data point. Jointly, the $n_2$ latent dimensions capture the marginal distribution of the input data patterns conditioned on the marginal embedding features. This results in disentangling of the original input data into individual factors of variation, where data presented in this univariate form would be more meaningful for visualization and analysis from a user perspective.

Furthermore, substituting the $D_{KL}$ term with the cross entropy loss eliminates the need for manual crafting of a prior, especially considering the case of a mixture model. The usual approach of a standard normal prior centers the latent embeddings at the origin. Alternatively, allowing $K$ classes and specifying $K$ means or centroids is also undesirable as this may inhibit the emergence of patterns inherent in the data. Moreover, while the $D_{KL}$ can be calculated for a Normal prior, this term is intractable when the probability distribution is a mixture model. In contrast, learning the parameters

that define the prior of each class in the mixture would allow the model to better capture and reflect relationships between and within classes.

A pictorial representation of DeepDIVE is summarized in Fig. 1.

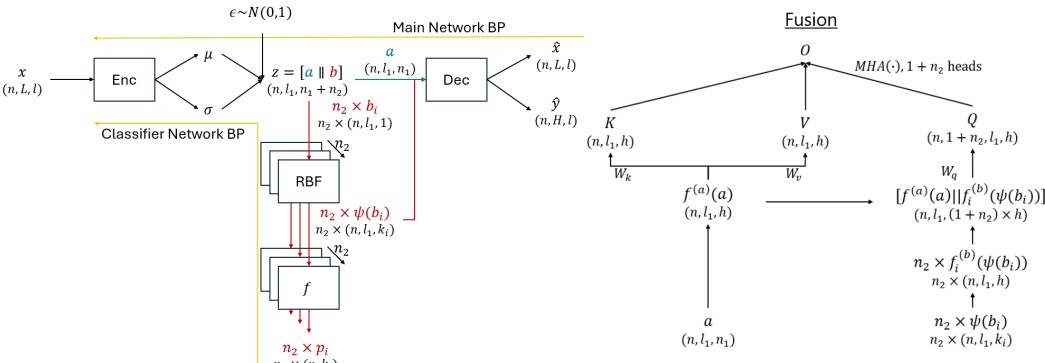

Figure 1: Model architecture for DeepDIVE.

In line with our derivations, our model individually tunes the $n_2$ marginal dimensions and $n_1$ conditional dimensions in an interleaving manner, with the the objective function being either $\mathcal{L}_{CE}$ for the marginal dimensions or $\mathcal{L}_{total}$ for the conditional dimensions. Formally, for the main network backpropagation, we minimize $\mathcal{L}_{total} = \mathcal{L}_{MSE}(\phi, \theta; x) + \mathcal{L}_{MSE}(\phi, \theta; y) + D_{KL}(\phi, \theta; a)$. For the classifier network backpropagation for label $i \in 1, \ldots, n_2$, we minimize $\mathcal{L}_{CE}(\phi, \theta; j_i)$. During each forward pass, noise is added either to only the relevant marginal embedding feature during each classification network pass, or to the conditional embedding features during the main network pass. To allow the model to learn conditional dimensions $a$ conditioned on marginal dimensions $b$, during main network backpropagation we freeze the RBF layers for the marginal dimensions, together with the weights in the last layer of the encoder that affect the marginal latent dimensions. The key here is to decouple the learning process of the marginal and conditional latent dimensions in an interleaving training scheme. Intuitively, this is similar to the alternating least squares algorithm (Zachariah et al., 2012), where the algorithm alternates between fixing the first factor when updating the second, and fixing the second factor when updating the first. However, one main difference is that in our case it is always the marginal dimensions that are fixed when the conditional dimensions are trained, so that it is always the conditional dimensions that are conditioned on the marginal ones.

To combine the information embedded by these separate marginal and conditional dimensions, we employ cross-attention in the fusion stage placed at the first layer of the decoder, directly after the RBF layer. Fig. 1 shows the basic idea for our proposed fusion stage, which involves cross-attention with both the conditional dimension and RBF outputs as the query, and the conditional dimensions as the key and value. For completeness, here we also state the expression for the attention mechanism proposed by Vaswani (2017):

$$\text{Attention} = \text{softmax}\left(\frac{QK^T}{\sqrt{h}}\right)V$$

so that multi-head attention (MHA) involves applying the attention function on submatrices of $Q$, $K$, and $V$ before concatenating the outputs. $f^{(a)}$ and $f^{(b)} = [f_1^{(b)}, \ldots, f_{n_2}^{(b)}]$ in the figure are projection matrices to align the dimensions of the conditional and marginal dimensions respectively. For our implementation, we have only used a simple multi-layer perceptron (MLP) as the remainder of the decoder after the fusion stage, but we would like to clarify that similar to the original VAE, the architecture within the encoder and decoder are flexible.

## 5  RESULTS

In this section, we empirically demonstrate the advantages of DeepDIVE on two time-series datasets: the gait dataset, for which assumption 2 does not hold and the electricity dataset, for which assumption 2 does hold. Our objective is to (i) highlight that despite being trained on multiple objective functions, our shared encoder trained on these multiple objectives is able to capture the posterior

distribution, and (ii) emphasize the benefit of disentangling the original input data into individual factors of variation.

*gait* (Zhang et al., 2023): Gait parameters for normal and pathological (Stroke, Parkinson's) walking patterns. This dataset contains no missing values and consists $l = 6$ readings from accelerometers and gyroscopes, with other information such as gait type and stride length.

For this dataset, we use input window size 1000, prediction window size 800, gap size 0 and split ratio 8:1:1. Our $n_2 = 2$ marginal dimensions correspond to gait type (Normal, Stroke, Parkinson's) and binned stride length (integer values from 0 to 13). Fig. 2 shows the correlation between gait type and stride length. Specifically, subjects with normal gait types have the longest stride lengths, while subjects with Parkinsons' have the shortest.

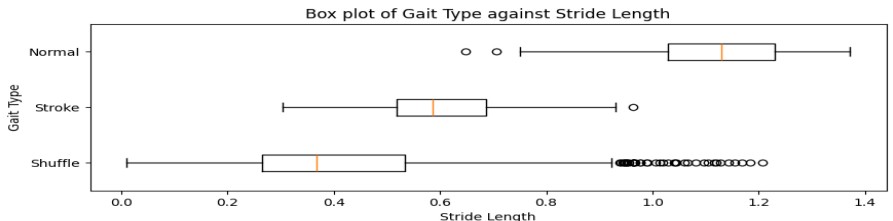

Figure 2: Correlation between Gait Type and Stride Length in *gait*.

*Electricity* (Lai et al., 2018): Cleaned and processed electricity consumption data, originally from Trindade (2015). Processed data contains no missing values and consists electricity consumption from $l = 321$ households in 1-hour windows.

For this dataset, we use input window size 168, prediction window size 1 at horizon 24, gap size 0 and split ratio 3:1:1. Our $n_2 = 3$ marginal dimensions correspond to hour of day, month, and day of week. These features are similar to discretized positional encoding with varying frequencies, which can be used if temporal feaures are unavailable.

For both datasets, we consider reconstruction and forecast accuracy using root relative squared error (RRSE), and we also report the standard deviation (std) of the RRSE across 30 runs. For fairness of comparison, we use the same encoder and decoder that DeepDIVE uses in implementing the baselines. We test DeepDIVE against the following baselines:

- DeepDIVE - $(a)$: DeepDIVE with only the conditional dimensions. This is equivalent to a VAE but with an additional forecasting branch.
- DeepDIVE - $(b)$: DeepDIVE with only the marginal dimensions.
- $\beta$-TCVAE: Same as DeepDIVE - $(a)$ except trained on the modified evidence lower bound proposed by Chen et al. (2018).

We chose VAE variants as baselines as we consider our proposed framework to have a strong theoretical foundation that is deeply connected to the VAE architecture. Since Proposition 1 decomposes the joint log-likelihood, this justifies the extension of a VAE for a forecasting task. Additionally, similar to DeepDIVE, the above baselines only maximise a lower bound on the joint log-likelihood, while other forecasting frameworks directly optimize the actual objective.

Tables 1 and 2 show that DeepDIVE achieves lower RRSE compared to our baselines, on the gait and electricity dataset respectively.

Fig. 3 illustrates the disentanglement in the representation space encoded by DeepDIVE for the electricity dataset. Each point represents the code $z = [\, a \parallel b \,] \in \mathbb{R}^{l \times (n_1 + n_2)}$ for a data sample, where $a \in \mathbb{R}^{l \times n_1}, b \in \mathbb{R}^{l \times n_2}$. For visualization on a 2-D diagram, the value on the y-axis is computed as the sum of the values across the $l$ household dimensions and $n_1$ conditional dimensions, ie. $y = \sum_{i=1}^{n_i} \sum_{j=1}^{l} a_{ji}$. Since the sum of Gaussian random variables is also Gaussian, this value is expected to also be Gaussian due to the $D_{KL}$ with Gaussian prior on the conditional dimensions. For marginal dimension $b_i$, the value on the x-axis is computed as a weighted sum of the values in $b_i$, using the weights for label $i$, denoted $w_i$ from the learned classifier network, ie. $x = \sum_{j=1}^{l} (w_i)_j (b_i)_j, w_i \in \mathbb{R}^l$. Each point is colored by its true class label $j$.

Table 1: Accuracy comparison for gait dataset across 30 runs

|  | Reconstruction | | Forecasting | | Disentanglement |
| --- | --- | --- | --- | --- | --- |
| Model | RRSE | std | RRSE | std | Mutual Information Gap (MIG) |
| DeepDIVE | 11.1627 | 4.8e-2 | 16.0582 | 3.7e-2 | 0.0473 |
| DeepDIVE - $(a)$ | 11.8835 | 3.1e-2 | 16.4434 | 3.8e-2 | 0.0155 |
| DeepDIVE - $(b)$ | 28.2268 | 0 | 27.7309 | 0 | 0.0464 |
| $\beta$-TCVAE | 29.3563 | 1.3e-5 | 33.7654 | 2.2e-5 | 0.0081 |

Table 2: Accuracy comparison for electricity dataset across 30 runs

|  | Reconstruction | | Forecasting | |
| --- | --- | --- | --- | --- |
| Model | RRSE | std | RRSE | std |
| DeepDIVE | 1.5803 | 3.6e-4 | 0.0998 | 7.1e-5 |
| DeepDIVE - $(a)$ | 2.5562 | 2.0e-3 | 0.1062 | 1.8e-5 |
| DeepDIVE - $(b)$ | 7.4779 | 0 | 0.1048 | 0 |
| $\beta$-TCVAE | 8.6409 | 0 | 0.1048 | 1.6e-8 |

Interestingingly, for the scatter plot of the aggregated conditional dimensions against the day marginal dimension, we observe a shift in distribution between data points encoded with $y < 600$ and the ones encoded with $y > 600$, more prominent on weekdays as compared to weekends. Further, comparing against the same plot for hour, we observe that these values of $y$ aligns with the subset of hours in the day from 12 to 6am. These observations align with our understanding of the variation in human activities and thus household electricity consumption patterns between weekdays and weekends, verifiable with household occupancy data. These results indicate that the learnt latent space have relevance to conditions in reality, thereby validating the capability of our model in effective learning of a representative latent data space in a format that facilitates analysis.

Unlike attention maps and convolutional neural network (CNN) feature maps, DeepDIVE presents data representations in a univariate format that easily lends itself to visualization and analysis by rendering it into a more high-level abstraction, thereby enhancing its usefulness for human-interpretability. Additionally, by encoding the categorical variates independently, the disentanglement also offers an avenue to test how these variables affect the reconstruction and forecasting tasks. While other popular methods such as Shapley values and decision trees may provide a more direct scoring approach to determine feature importance, DeepDIVE offers insights into how these categorical features are encoded in relation to the downstream tasks.

# 6 CONCLUSION

In this paper, we present a theoretical extension of the VAE to accommodate forecasting scenarios, with disentanglement via the application of Bayes' theorem with the "naïve" assumption of independence among the supervised dimensions conditional on the original input. Despite being multi-objective in nature, our overall loss function was derived from the log likelihood of a pre-existing data point, hence it follows that these objectives exhibit no mutual conflict. Leveraging this mathematical basis, we present Deep Disentangled Interleaving Variational Encoding (Deep-DIVE), designed for learning a disentangled representation space without conflicting objectives in interleaving multi-task learning. Experimental validation across 2 datasets confirms the benefits of DeepDIVE, which utilizes the shared representation space to achieve results superior to both the original VAE and $\beta$-TCVAE, and comparable to existing state-of-the-art. Exploring the capabilities of the shared representation space encoded by DeepDIVE in other downstream tasks such as anomaly detection and augmenting large language models (LLMs) offers a compelling direction for future research.

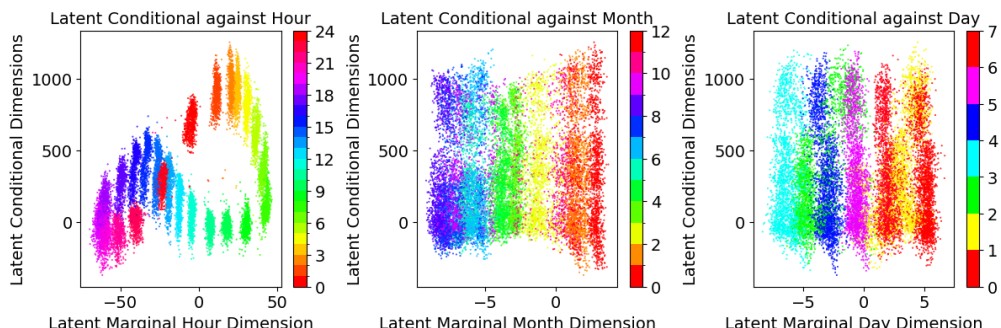

Figure 3: Disentangled representation space for *electricity*.

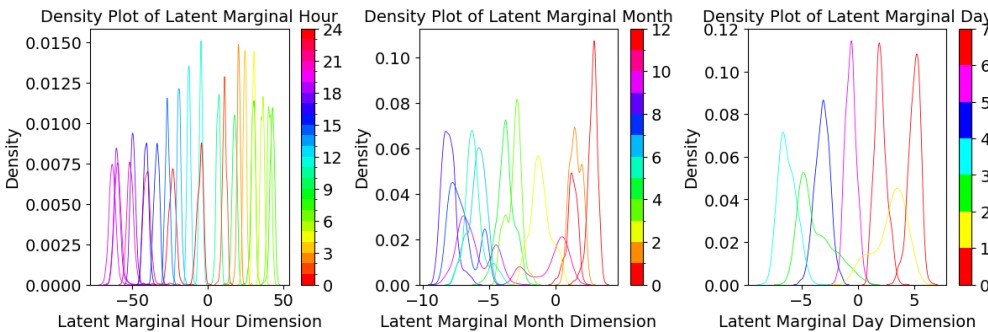

Figure 4: Density of the latent embeddings along each marginal dimension of representation space for *electricity*. Compared to Fig. 3, in which there may be overlaps, Fig. 4 more clearly shows the distribution and concentration of data points along the marginal dimensions, for easier identification of class distributions along each dimension.

## 7 REPRODUCIBILITY AND ETHICS STATEMENT

This paper presents work which aims to advance the mathematical understanding in the field of machine learning. We acknowledge that all authors have read the Code of Ethics and pledge to adhere to its principles and guidelines in our research work. Given that our study is done on a generative model, we have tried our best to ensure a certain degree of reproducibility of results, although results may not be exactly reproducible.

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

# A APPENDIX

## A.1 PROBABILISTIC CORRELATED TIME-SERIES FORECASTING

Probabilistic time-series forecasting aims to learn the correlation between historical signals and future outcomes to obtain a probability distribution over possible future outcomes. The classical methods for probabilistic multivariate forecasting are mainly dedicated to AR models, which can be represented in state space form, and Bayesian forecasting methods (Rangapuram et al., 2018; Qiu et al., 2018; Dorfman & Havenner, 1992). In more recent years, the advancement of data collection and increasing power of computing facilities have increased the feasibility of deep networks, which have gained popularity in multivariate probabilistic time-series forecasting due to their ability to capture non-linear temporal dependencies in the data. For example, DeepAR (Salinas et al., 2020) is a popular univariate forecasting method which fits a shared global RNN across multiple related time-series after scaling, where the network is adapted to probabilistic forecasting by Monte Carlo sampling methods. Closer to our work, the Temporal Latent Autoencoder (TLAE) (Nguyen & Quanz, 2021) enhances the DeepGLO (Sen et al., 2019), which uses low-rank matrix factorization to obtain a set of basis time-series that captures global properties, and a Temporal Convolutional Network (TCN) on these basis time-series to generate global forecasts. The global forecasts are then concatenated with other variables and fed into another TCN, viewed as the local TCN, to forecast for each individual time-series. Building on this study, TLAE replaces the global TCN with an encoder and local TCN with a decoder in an autoencoder architecture, with a temporal deep neural network model in the latent space to encourage evolution of the embeddings over time. Since TLAE introduces non-linearity into the global encoder and local decoder and is expected to perform better in cases when there are sufficient training data, from among this group we only compare against TLAE in our experiments.

However, we note that the use of an RNN within the latent space may overly reduce the flexibility of the model, since the temporal trends are learnt within the bottleneck section of the model. With a similar idea of global and local forecasting, DSANet (Huang et al., 2019) uses a dual-branch TCN where one branch operates on a global level and the other operates on a local level, before fusing the features from both branches with a fully connected MLP to capture complex linear combinations of both global and local temporal patterns. The authors of AutoCTS (Wu et al., 2021) introduce a design with both micro and macro search spaces that model possible architectures, and uses Neural Architecture Search (NAS) (Elsken et al., 2019) to jointly explore the search spaces and automatically discover highly competitive models. Models identified by the framework has been shown to outperform state-of-the-art human-designed models, thus eliminating the need for manual design. LightCTS (Lai et al., 2023), with its adoption of plain stacking of TCN and transformer, can be viewed as an autoencoder with a TCN encoder and transformer decoder and an interpretable latent space, that is capable of state-of-the-art accuracies.

Compared to these baselines, we consider forecast accuracy using RRSE on the electricity dataset, and we also report the standard deviation of the RRSE across 30 runs. Table 3 shows that despite being trained on multiple objective functions, derived from the loss function $\mathcal{L}(\theta, \phi; x, y)$ that is only a lower bound of the true negative log-likelihood loss $\log p_\theta(x, y)$, DeepDIVE still produces results comparable to the existing state-of-the-art trained on only the forecasting task.

Table 3: Accuracy comparison for electricity dataset across 30 runs

| Model | 3-th RRSE | std | 6-th RRSE | std | 12-th RRSE | std | 24-th RRSE | std |
|---|---|---|---|---|---|---|---|---|
| TLAE | 0.5633 | 3.6e-4 | 0.5629 | 1.2e-4 | 0.5636 | 4.5e-4 | 0.5632 | 1.8e-7 |
| DsaNet | 0.0855 | 0 | 0.0963 | 0 | 0.1020 | 0 | 0.1044 | 0 |
| AutoCTS | 0.0743 | 0 | 0.0865 | 0 | 0.0932 | 0 | 0.0947 | 0 |
| AutoCTS-KDF | 0.0818 | 0 | 0.0949 | 0 | 0.1003 | 0 | 0.1018 | 0 |
| AutoCTS-KDP | 0.0764 | 0 | 0.0899 | 0 | 0.0934 | 0 | 0.0983 | 0 |
| LightCTS | 0.0736 | 0 | 0.0831 | 0 | 0.0898 | 0 | 0.0952 | 0 |
| DeepDIVE | 0.0887 | 6.0e-6 | 0.0911 | 1.0e-5 | 0.0995 | 7.1e-5 | 1.0000 | 8.1e-5 |

## A.2 TECHNICAL PROOFS

### A.2.1 PROOF OF PROPOSITION 1

$$\log p_\theta(x,y) = \int_A \int_B q_\phi(a,b|x) \log p_\theta(x,y) \, db \, da \tag{34}$$

$$= \mathbb{E}_{A,B \sim q_\phi}[\log p_\theta(x,y)] \tag{35}$$

$$= \mathbb{E}_{A,B \sim q_\phi}\left[\log \frac{p_\theta(a,b,x,y)}{p_\theta(a,b|x,y)}\right] \tag{36}$$

$$= \mathbb{E}_{A,B \sim q_\phi}\left[\log \left(\frac{p_\theta(a,b,x,y)}{q_\phi(a,b|x)} \frac{q_\phi(a,b|x)}{p_\theta(a,b|x,y)}\right)\right] \tag{37}$$

$$= \mathbb{E}_{A,B \sim q_\phi}\left[\log \frac{p_\theta(a,b,x,y)}{q_\phi(a,b|x)}\right] + \mathbb{E}_{A,B \sim q_\phi}\left[\log \frac{q_\phi(a,b|x)}{p_\theta(a,b|x,y)}\right] \tag{38}$$

$$=: \mathcal{L}(\theta,\phi;x,y) + D_{KL}(q_\phi(a,b|x) \parallel p_\theta(a,b|x,y)) \tag{39}$$

where

$$\mathcal{L}(\theta,\phi;x,y) = \mathbb{E}_{A,B \sim q_\phi}\left[\log \frac{p_\theta(a,b,x,y)}{q_\phi(a,b|x)}\right] \tag{40}$$

$$= \mathbb{E}_{A,B \sim q_\phi}[\log p_\theta(y|a,b,x)] + \mathbb{E}_{A,B \sim q_\phi}[\log p_\theta(x|a,b)] + \mathbb{E}_{A,B \sim q_\phi}\left[\log \frac{p_\theta(a,b)}{q_\phi(a,b|x)}\right] \tag{41}$$

$$=: \text{forecast loss} + \text{reconstruction loss} - D_{KL}(q_\phi(a,b|x) \parallel p_\theta(a,b)) \tag{42}$$

### A.2.2 PROOF OF PROPOSITION 2

$$D_{KL}(q_\phi(a,b|x) \parallel p_\theta(a,b)) \tag{43}$$

$$= \int_A \int_B q_\phi(a,b|x) \log \frac{q_\phi(a,b|x)}{p_\theta(a,b)} \, db \, da \tag{44}$$

$$= \int_A \int_B q_\phi(a|b,x) q_\phi(b|x) \log \frac{q_\phi(a|b,x)}{p_\theta(a)} \frac{q_\phi(b|x)}{p_\theta(b)} \, db \, da \tag{45}$$

$$= \int_A \int_B q_\phi(a|b,x) q_\phi(b|x) \log \frac{q_\phi(a|b,x)}{p_\theta(a)} \, db \, da + \int_A \int_B q_\phi(a|b,x) q_\phi(b|x) \log \frac{q_\phi(b|x)}{p_\theta(b)} \, db \, da \tag{46}$$

$$= \int_B q_\phi(b|x) \int_A q_\phi(a|b,x) \log \frac{q_\phi(a|b,x)}{p_\theta(a)} \, da \, db + \int_B q_\phi(b|x) \log \frac{q_\phi(b|x)}{p_\theta(b)} \int_A q_\phi(a|b,x) \, da \, db \tag{47}$$

$$= \mathbb{E}_{B \sim q_\phi}[D_{KL}(q_\phi(a|b,x) \parallel p_\theta(a))] + D_{KL}(q_\phi(b|x) \parallel p_\theta(b)) \tag{48}$$

where the second equality is due to assumption 1.

### A.2.3 PROOF OF PROPOSITION 3

$$D_{KL}(q_\phi(b_i, b_j|x) \| p_\theta(b_i, b_j)) \tag{49}$$

$$= \int_{B_i} \int_{B_j} q_\phi(b_i, b_j|x) \log \frac{q_\phi(b_i, b_j|x)}{p_\theta(b_i, b_j)} \, db_j \, db_i \tag{50}$$

$$= \int_{B_i} \int_{B_j} q_\phi(b_i|x)q_\phi(b_j|x) \log \frac{q_\phi(b_i|x)q_\phi(b_j|x)}{p_\theta(b_i)p_\theta(b_j)} \, db_j \, db_i \tag{51}$$

$$= \int_{B_i} \int_{B_j} q_\phi(b_i|x)q_\phi(b_j|x) \log \frac{q_\phi(b_i|x)}{p_\theta(b_i)} \, db_j \, db_i + \int_{B_i} \int_{B_j} q_\phi(b_i|x)q_\phi(b_j|x) \log \frac{q_\phi(b_j|x)}{p_\theta(b_j)} \, db_j \, db_i \tag{52}$$

$$= D_{KL}(q_\phi(b_i|x) \| p_\theta(b_i)) + D_{KL}(q_\phi(b_j|x) \| p_\theta(b_j)) \tag{53}$$

where the second equality is due to assumption 2.

### A.2.4 PROOF OF PROPOSITION 4

$$D_{KL}(q_\phi(b|x) \| p_\theta(b)) \tag{54}$$

$$= \int_B q_\phi(b|x) \log \frac{q_\phi(b|x)}{p_\theta(b)} \, db \tag{55}$$

$$= \int_B q_\phi(b|x) \log q_\phi(b|x) \, db - \int_B q_\phi(b|x) \log \sum_{k=1}^{K} p_\theta(b, k) \, db \tag{56}$$

where the first term of the last line is equal to $-H(B|X = x)$ by definition of conditional differential entropy. From an information theoretic perspective, maximizing the entropy of the encoder output increases the information gain.

For the second term,

$$- \int_B q_\phi(b|x) \log \sum_{k=1}^{K} p_\theta(b, k) \, db \tag{57}$$

$$= \mathbb{E}_{B \sim q_\phi} \left[ - \log \sum_{k=1}^{K} p_\theta(b, k) \right] \tag{58}$$

$$= \mathbb{E}_{B \sim q_\phi} \left[ - \log \sum_{k=1}^{K} Q(k) \frac{p_\theta(b, k)}{Q(k)} \right] \text{ for any } Q(k) \text{ st. } 0 < Q(k) \le 1 \,\forall\, k \in \{1, ..., K\}, \sum_{k=1}^{K} Q(k) = 1 \tag{59}$$

$$\le \mathbb{E}_{B \sim q_\phi} \left[ \sum_{k=1}^{K} Q(k) \left[ - \log \frac{p_\theta(b, k)}{Q(k)} \right] \right] \text{ by Jensen's inequality, where the inequality is tight if } Q(k) = p_\theta(k|b) \tag{60}$$

### A.2.5 PROOF OF PROPOSITION 5

In this proof we use the fact that the global minimum on a function differentiable everywhere implies that the derivative is 0.

Consider

$$Q(k) = p_{\theta_t}(k|b)$$

(Necessity) Then the value of $\theta$ which minimizes the upper bound

$$\theta_{t+1} = \arg\min_\theta \mathbb{E}_{B\sim q_\phi}\left[\sum_{k=1}^K p_{\theta_t}(k|b)\left[-\log\frac{p_\theta(b,k)}{p_{\theta_t}(k|b)}\right]\right] \tag{61}$$

$$= \arg\min_\theta \mathbb{E}_{B\sim q_\phi}\left[\sum_{k=1}^K p_{\theta_t}(k|b)[-\log p_\theta(b,k) + \log p_{\theta_t}(k|b)]\right] \tag{62}$$

$$= \arg\min_\theta \mathbb{E}_{B\sim q_\phi}\left[-\sum_{k=1}^K p_{\theta_t}(k|b)\log p_\theta(b,k) + \sum_{k=1}^K p_{\theta_t}(k|b)\log p_{\theta_t}(k|b)\right] \tag{63}$$

$$= \arg\min_\theta \mathbb{E}_{B\sim q_\phi}\left[-\sum_{k=1}^K p_{\theta_t}(k|b)\log p_\theta(b,k)\right] + \mathbb{E}_{B\sim q_\phi}\left[\sum_{k=1}^K p_{\theta_t}(k|b)\log p_{\theta_t}(k|b)\right] \tag{64}$$

occurs at the point where

$$\frac{\partial}{\partial\nu}\mathbb{E}_{B\sim q_\phi}\left[-\sum_{k=1}^K p_{\theta_t}(k|b)\log p_\theta(b,k)\right] = 0 \text{ and } \frac{\partial}{\partial\tau}\mathbb{E}_{B\sim q_\phi}\left[-\sum_{k=1}^K p_{\theta_t}(k|b)\log p_\theta(b,k)\right] = 0 \tag{65}$$

and thus

$$\mathbb{E}_{B\sim q_\phi}\left[-\sum_{k=1}^K p_{\theta_t}(k|b)\frac{\partial}{\partial\nu}\log p_\theta(b,k)\right] = 0 \text{ and } \mathbb{E}_{B\sim q_\phi}\left[-\sum_{k=1}^K p_{\theta_t}(k|b)\frac{\partial}{\partial\tau}\log p_\theta(b,k)\right] = 0 \tag{66}$$

since functions parameterized by $\phi$ and $\theta_t$ are constant with respect to $\theta$. Differentiating the log term in the above expression, we have

$$\mathbb{E}_{B\sim q_\phi}\left[-\sum_{k=1}^K p_{\theta_t}(k|b)\left(\frac{\frac{\partial}{\partial\nu}p_\theta(b,k)}{p_\theta(b,k)}\right)\right] = 0 \text{ and } \mathbb{E}_{B\sim q_\phi}\left[-\sum_{k=1}^K p_{\theta_t}(k|b)\left(\frac{\frac{\partial}{\partial\tau}p_\theta(b,k)}{p_\theta(b,k)}\right)\right] = 0 \tag{67}$$

which, by chain rule, further evaluates to

$$\mathbb{E}_{B\sim q_\phi}\left[-\sum_{k=1}^K p_{\theta_t}(k|b)\left(\frac{\frac{\partial}{\partial\nu}p_\theta(b|k)p_\theta(k)}{p_\theta(b,k)}\right)\right] = 0 \text{ and } \mathbb{E}_{B\sim q_\phi}\left[-\sum_{k=1}^K p_{\theta_t}(k|b)\left(\frac{\frac{\partial}{\partial\tau}p_\theta(b|k)p_\theta(k)}{p_\theta(b,k)}\right)\right] = 0 \tag{68}$$

By assumption 3, any $\theta^*$ that minimizes the upper bound must satisfy the following equations

$$\mathbb{E}_{B\sim q_\phi}\left[-\sum_{k=1}^K p_{\theta_t}(k|b)\left(\frac{p_\theta(k)}{p_\theta(b,k)}\frac{\partial}{\partial\nu}\psi_k(b)\right)\right] = 0 \tag{69}$$

$$\mathbb{E}_{B\sim q_\phi}\left[-\sum_{k=1}^K p_{\theta_t}(k|b)\left(\frac{p_\theta(k)}{p_\theta(b,k)}\frac{\partial}{\partial\tau}\psi_k(b)\right)\right] = 0 \tag{70}$$

### A.2.6 PROOF OF SUFFICIENCY COROLLARY

Observe that many common probability distributions are log-concave in their parameters (eg. $N(\mu,\sigma^2)$ in $\mu$, $\text{Exp}(\lambda)$ in $\lambda$), or have the stationary point where likelihood is maximum (eg. $N(\mu,\sigma^2)$ in $\sigma$). Thus, if $p_\theta(b|k)$ is chosen such that:

- $p_\theta(b|k)$ is a valid probability distribution
- $p_\theta(b|k)$ is log-concave in its parameters

then the upper bound

$$\mathbb{E}_{B\sim q_\phi}\left[\sum_{k=1}^{K} Q(k)\left[-\log\frac{p_\theta(b,k)}{Q(k)}\right]\right] \tag{71}$$

$$=\mathbb{E}_{B\sim q_\phi}\left[\sum_{k=1}^{K} Q(k)\left[-\log\frac{p_\theta(b|k)p_\theta(k)}{Q(k)}\right]\right] \tag{72}$$

$$=\mathbb{E}_{B\sim q_\phi}\left[\sum_{k=1}^{K} Q(k)\left[-\log\frac{p_\theta(k)}{Q(k)}\right]\right]+\mathbb{E}_{B\sim q_\phi}\left[\sum_{k=1}^{K} Q(k)\left[-\log p_\theta(b|k)\right]\right] \tag{73}$$

has $p_\theta(k)\approx\frac{n_k}{n}$ by assumption 4 and second RHS term convex by our choice of $p_\theta(b|k)$, since the sum of convex functions is convex. Thus the necessary conditions equation 25 and equation 26 become sufficient conditions for optimality, since this implies that the stationary point of the upper bound is a global minimum point.

### A.2.7 PROOF OF PROPOSITION 6

Similar to the proof in Appendix A.2.5, in this proof we also use the fact that the global minimum on a function differentiable everywhere implies that the derivative is 0.

If there exists a point $\theta^*$ that minimizes the cross entropy loss for each class $k \in 1, \ldots, K$ respectively, where the cross entropy loss is as defined in definition 1, then this point must satisfy

$$\frac{\partial}{\partial\nu}-\log p_\theta(j|b)=0 \text{ and } \frac{\partial}{\partial\tau}-\log p_\theta(j|b)=0 \text{ and } \frac{\partial}{\partial\theta-\{\nu,\tau\}}-\log p_\theta(j|b)=0 \tag{74}$$

$$\frac{1}{p_\theta(j|b)}\frac{\partial}{\partial\nu}p_\theta(j|b)=0 \text{ and } \frac{1}{p_\theta(j|b)}\frac{\partial}{\partial\tau}p_\theta(j|b)=0 \text{ and } \frac{1}{p_\theta(j|b)}\frac{\partial}{\partial\theta-\{\nu,\tau\}}p_\theta(j|b)=0 \tag{75}$$

We first prove the case for 2 classes, where by definition of the sigmoid function we have $p_\theta(j|b)=\sigma(f(\boldsymbol{\psi}(b)))=\frac{1}{e^{-f(\boldsymbol{\psi}(b))}+1}$. Observe also that the softmax function in the 2 class case $\frac{e^{y_0}}{e^{y_0}+e^{y_1}}$ reduces to the sigmoid if $y_0$ is fixed at 0. Then

$$\frac{\partial}{\partial\nu}\sigma(f(\boldsymbol{\psi}(b)))=0 \text{ and } \frac{\partial}{\partial\tau}\sigma(f(\boldsymbol{\psi}(b)))=0 \text{ and } \frac{\partial}{\partial\theta-\{\nu,\tau\}}\sigma(f(\boldsymbol{\psi}(b)))=0 \tag{76}$$

Thus any $\theta^*$ that minimizes the cross entropy loss satisfies

$$\sigma(f(\boldsymbol{\psi}(b)))[1-\sigma(f(\boldsymbol{\psi}(b)))]\frac{\partial}{\partial\psi}f(\boldsymbol{\psi}(b))\frac{\partial}{\partial\nu}\boldsymbol{\psi}(b)=0 \tag{77}$$

$$\sigma(f(\boldsymbol{\psi}(b)))[1-\sigma(f(\boldsymbol{\psi}(b)))]\frac{\partial}{\partial\psi}f(\boldsymbol{\psi}(b))\frac{\partial}{\partial\tau}\boldsymbol{\psi}(b)=0 \tag{78}$$

$$\sigma(f(\boldsymbol{\psi}(b)))[1-\sigma(f(\boldsymbol{\psi}(b)))]\frac{\partial}{\partial\theta-\{\nu,\tau\}}f(\boldsymbol{\psi}(b))=0 \tag{79}$$

Finally, note that by property of the sigmoid function, $0<\sigma(x)<1\ \forall\ x\in\mathbb{R}$, hence $\sigma(x)\neq 0$ and $1-\sigma(x)\neq 0\ \forall\ x\in\mathbb{R}$.

Additionally, if $\psi_k(b)$ and $f(x)$ are selected such that $\alpha\leq\psi_k(b)\leq\beta\ \forall\ b\in\mathbb{R}, k\in\{1,...,K\}$ and $\frac{\partial}{\partial x}f(x)\neq 0\ \forall\ x\in[\alpha,\beta]$ then equation 77 and equation 78 imply that

$$\frac{\partial}{\partial\nu}\boldsymbol{\psi}(b)=\mathbf{0} \tag{80}$$

$$\frac{\partial}{\partial\tau}\boldsymbol{\psi}(b)=\mathbf{0} \tag{81}$$

### A.2.8 GRAPHICAL OVERVIEW OF LOSS FUNCTION DERIVATION

Fig. 5 is a graphical illustration of the role played by some of the key terms and assumptions in the derivations, which readers may find helpful as an overview.

$$\log p_\theta(x, y) =: \underline{\mathcal{L}(\theta, \phi; x, y)} + D_{KL}(q_\phi(a, b|x) \parallel p_\theta(a, b|x, y))$$

$$\parallel$$

Forecast + Reconstruction $- \underline{D_{KL}(q_\phi(a, b|x) \parallel p_\theta(a, b))}$

Assumption 1 allows separation between conditional and marginal dimensions in prior

$$- \mathbb{E}_{B \sim q_\phi}[D_{KL}(q_\phi(a|b, x) \parallel p_\theta(a))] + \underline{\text{Total marginal KL-divergence}}$$

Assumption 1 allows separation among marginal dimensions in prior
Assumption 2 allows separation among marginal dimensions in probabilistic encoder

$- \underline{\text{Sum of divergences for individual marginal dimensions}}$

Assumption 3 needed for log-concavity of probability distribution
Assumption 4 allows prior probability of the class to be independent of kernel parameters

Common maximizer

$$\parallel$$

$- $ Cross Entropy

Figure 5: Graphical overview of loss function derivation for DeepDIVE, with corresponding assumptions made.

