# OpenReview forum: "Disentangled interleaving variational encoding"
_ICLR.cc/2025/Conference — Submitted to ICLR 2025_

### Official Review · Reviewer_N7k7 · 2024-10-25

**Soundness:** 2
**Presentation:** 1
**Contribution:** 2
**Rating:** 3
**Confidence:** 3

**Summary:**

This paper proposes the new VAE architecture that has a special latent variable structure for time series forecasting.
This paper assumes that the target variable of VAE is a combination of future and past variables,
and that the future variables do not depend on the latent variables, but only on the past variables.
The latent variables also consist of two vectors $a$ and $b$, and the posterior distribution of $a$ depends on $b$, but that of $b$ is independent of $a$.
Furthermore, it is assumed that $b$ consists of a mixture model of K classes. Finally, the likelihood function obtained under these assumptions is the proposed objective function.
In the experiment, the forecasting accuracy is compared using two time series datasets, and the visualization of the latent variable is discussed.

**Strengths:**

- The proposed method is well explained using mathematical equations.
The expansion of equations is easy to understand, and the assumptions necessary for the expansions are also clearly stated.
However, as written in Weakness, the practical validity and motivation of the assumptions are unclear.
- In the experiments, the forecasting accuracy is higher than that of VAE and beta VAE on one dataset.
However, if you only focus on the forecasting accuracy, it is difficult to say that VAE and beta VAE are appropriate baselines.
If the paper is a study of application, it should be compared with baselines for time series forecasting.
On the other dataset, baseline methods seem to be forecasting methods, but the proposed method does not necessarily outperform these baselines.

**Weaknesses:**

- This paper does not succeed in positioning itself in the context of previous research.
I strongly suggest that this paper should clarify whether the focus of the paper is a proposal of a method for a specific application or one of a fundamental method.
The abstract seems to state that the paper solves a problem in general multi-task learning, but the introduction seems to claim that it is a technique for solving the problem of time series (or ASC?) forecasting.
It is inconsistent and makes the motivation unclear. The paper needs to consistently state the research problem and clearly explain why the proposed method is effective for that problem.
If this paper solves a specific problem in time series forecasting, it should justify why that problem is important and common to broad domains.

- Related to the above, it is difficult to understand the reasons and principles behind the design of the proposed method.
Although the assumptions that lead to the theorem are shown, this paper does not mention the assumptions of the target tasks or the most suitable use cases of the proposed method.
As a result, it is difficult to discuss the generality and importance of the task being solved by the proposed method, and the impact of this paper is unclear.
The graphical models of a, b, x, and y might be useful for readers to understand the assumptions,
and illustrations of how these variables relate to practical tasks will also help readers understand the effectiveness of the proposed method.

- The paper evaluates the visualization of latent variables but does not compare it with existing methods.
I think that disentanglement of latent variables in time-series data have already been discussed in previous research such as [a].
If the claim of this paper is the importance of disentangling latent variables, the proposed method should be compared with previous methods.

[a] Fraccaro, Marco, et al. "A disentangled recognition and nonlinear dynamics model for unsupervised learning." Advances in neural information processing systems 30 (2017).

- The baseline is not consistent in the evaluation on the two datasets.
Why are the baselines different in Table 1 and Table 2? Is it appropriate to compare VAE and beta VAE using time series data sets?

**Questions:**

If I have misunderstood the paper, please point out it.

---

> ### Author Response · Authors · 2024-11-25
> **Official Comment by Authors**
>
> We appreciate the reviewer’s recognition of our theoretical contributions and for highlighting a number of important points.
>
> We have changed the literature review and results section to enhance the presentation. Below are our responses to weaknesses and questions.
>
> > This paper assumes that the target variable of VAE is a combination of future and past variables, and that the future variables do not depend on the latent variables, but only on the past variables.
>
> If we could provide some clarification to this section, we note that the future variables do depend on both latent and past variables. This stems from Proposition 1, where the first term of the ELBO function is an expectation of the conditional log-likelihood of future variables conditioned on both the latent and past variables.
>
> **Addressing Weaknesses and Questions:**
> > This paper does not succeed in positioning itself in the context of previous research. I strongly suggest that this paper should clarify whether the focus of the paper is a proposal of a method for a specific application or one of a fundamental method. The abstract seems to state that the paper solves a problem in general multi-task learning, but the introduction seems to claim that it is a technique for solving the problem of time series (or ASC?) forecasting. It is inconsistent and makes the motivation unclear. The paper needs to consistently state the research problem and clearly explain why the proposed method is effective for that problem. If this paper solves a specific problem in time series forecasting, it should justify why that problem is important and common to broad domains.
>
> Thank you for highlighting this potential avenue for confusion. To clarify, we are looking to propose a generic theoretical framework, and we agree that this can be made clearer in the paper. We have also removed mentions of ASC and explainability in the paper, so as to maintain focus on the paper’s objective. As other reviewers have also raised similar questions regarding the research problem and motivation, we have addressed this issue in the general reply to all reviewers. We have also placed it here for the reviewer’s convenience.
>
> We motivate the use of semi-supervised learning [in DeepDIVE] considering the fact that while it is common to employ a sequence as input to a time series forecasting model, other available labels that classify the entire sequence are often overlooked. For this task, the benefits of our variational encoding architecture are two-fold. Firstly, building upon the statistical foundations provided by the VAE for learning deep latent-variable models and corresponding inference models for reconstruction, DeepDIVE also provides a principled and well-founded semi-supervised framework for learning deep latent-variable models and corresponding inference models which is useful for reconstruction, forecasting, and classification. Secondly, we believe that probabilistic latent variable models such as VAEs hold great potential for facilitating autonomous planning and resource allocation in situations characterized by uncertainty, mirroring real-world scenarios.
>
> We have modified the introduction to reflect this focus.
>
> > Related to the above, it is difficult to understand the reasons and principles behind the design of the proposed method. Although the assumptions that lead to the theorem are shown, this paper does not mention the assumptions of the target tasks or the most suitable use cases of the proposed method. As a result, it is difficult to discuss the generality and importance of the task being solved by the proposed method, and the impact of this paper is unclear. The graphical models of a, b, x, and y might be useful for readers to understand the assumptions, and illustrations of how these variables relate to practical tasks will also help readers understand the effectiveness of the proposed method.
>
> The principles behind the design of DeepDIVE stem from Theorem 1, which extends the ELBO function to also include losses for forecasting and classification, thus informing our design (addition of a forecasting branch, addition of a classification branch, employing interleaving training, partition of the latent space into marginal and conditional dimensions, integrating RBF). Additionally, regarding the reason and motivation for this framework, we place it in the response to the previous question, as they are related.
>
> [response continued in the next comment]

---

> > ### Author Response · Authors · 2024-11-25
> > **Official Comment by Authors**
> >
> > [response continued from the previous comment]
> >
> > Regarding assumptions on the target tasks, they are also based on the theoretical derivations and assumptions made. For example, this framework requires that historical variables and forecast variables are available for training, and that historical variables are available during inference. Regarding the labels that classify the entire sequence, these can be applied in a plug-and-play style, or removed if such labels are not available by setting the number of marginal dimensions $n_2$ to be zero. Hence we believe DeepDIVE is a very general framework that can be applied to existing autoencoder training pipelines for extension to forecasting use cases.
> >
> > Additionally, related to this point on the relation between assumptions made in theory and in practical tasks, since one of our main assumption is the Naïve conditional independence assumption in Assumption 2, which often does not hold true in practice, we have endeavored to understand/observe the extent to which the model’s performance is affected from a violation of this assumption. We show with experimental results that even on a dataset with clearly correlated classes, DeepDIVE performs better in terms of both forecast and reconstruction loss compared to the same architecture without the disentanglement, and provide an intuitive hypothesis as to why this might be the case in the remarks after Assumption 2 in the revised manuscript.
> >
> > We agree with the reviewer’s observation that readers may not find it easy to understand the effectiveness of our proposed method, and as such we have newly added a graphical model in Section 3 to illustrate the role played by some of the key terms and assumptions in the derivations.
> >
> > > The paper evaluates the visualization of latent variables but does not compare it with existing methods. I think that disentanglement of latent variables in time-series data have already been discussed in previous research such as [a]. If the claim of this paper is the importance of disentangling latent variables, the proposed method should be compared with previous methods.
> >
> > We felt that the proposed Kalman Variational Autoencoder (KVAE) is less suitable as a comparison for a couple of reasons. For a given series of length T, KVAE requires computation of the smoothed posterior $p_\gamma (z_t |a_{1:T},u_{1:T})$ on its backward pass, which assumes availability of both past and future information. The model is a data imputation method for handling missing data, and the authors have emphasised that the proposed KVAE focuses on inference, not prediction, which does not seem to align with DeepDIVE’s objective. Moreover, KVAE’s disentanglement refers to disentangling the model’s recognition of the object itself from temporal dynamics that govern the environment that the object is in, whereas in our case our derivations only involve the temporal dynamics, and we do not assume the presence of an object.
> >
> > However, we agree with the reviewer’s assessment that there should be a comparison against previous methods regarding the disentanglement in the latent space. We have newly implemented β-TCVAE, which numerical experiments suggest to be able to discover more interpretable representations compared to β-VAE. Since this is related to a suggestion by reviewer ETkb, we have addressed this issue there. We have also placed it here for the reviewer’s convenience.
> >
> > We have implemented β-TCVAE with λ=0, β=6, and parameters (`include_mutinfo=True`, `tcvae=True`). For fairness of comparison, we use the same encoder and decoder that DeepDIVE uses for the implementation of β-TCVAE, and have also added logpy to the modified ELBO function that β-TCVAE uses, where the distributions of x, y and the prior are defined as $N(\bar{x}, \sigma_x)$ and $N(\bar{y}, \sigma_y)$ respectively. Similar to the original study in the cited paper, we find that this compromises the density estimation, as the reconstruction and forecast RRSE of the model trained on the modified ELBO function exhibits lower performance compared to DeepDIVE. We have newly added these results to Section 5.
> >
> > [continued in the next comment]

---

> ### Author Response · Authors · 2024-11-25
> **Official Comment by Authors**
>
> [continued from the previous comment]
>
> > The baseline is not consistent in the evaluation on the two datasets. Why are the baselines different in Table 1 and Table 2? Is it appropriate to compare VAE and beta VAE using time series data sets?
>
> We respect the reviewer's perspective regarding the suitability of VAE and β-VAE as appropriate baselines. However, we would like to offer a different viewpoint. We consider our proposed framework to have a strong theoretical foundation that is deeply connected to the VAE architecture. In our opinion, the comparison against VAE and β-VAE should be deemed appropriate for a forecasting task as Proposition 1 decomposes the joint log-likelihood, which justifies the extension of a VAE for such a task. Furthermore, a comparison among these VAE variants may even be more suitable as, similar to DeepDIVE, they only maximise a lower bound on the joint log-likelihood, while other forecasting frameworks directly optimize the mean-squared error. In light of this, we have relocated the results regarding the forecasting baselines (Table 2) to the Appendix. However, we found it important to still include them (in the Appendix) to demonstrate that DeepDIVE can achieve forecasting results on par with existing state-of-the-art forecasting methods, even when trained on only a bound of the actual objective.

---

> > ### Comment · Reviewer_N7k7 · 2024-11-27
> > **Thank you for the feedback**
> >
> > Thank you for your rebuttal and revision.
> > I think the authors' efforts in a short time are great. However, the introduction is already more than 40% different, and the amount of revisions such as new figures and deleting sub-sections exceeds the level that can be judged during the discussion period. I think that it is necessary to be reviewed again from the beginning as a new submission.
> >
> > In the revision, Introduction is consistent with Abstract: they follow the story of general multi-task learning methods. However, from the rebuttal, it seems clear that time series forecasting is the motivation. For example, the rebuttal claims the following:
> > "We motivate the use of semi-supervised learning [in DeepDIVE] considering the fact that while it is common to employ a sequence as input to a time series forecasting model, other available labels that classify the entire sequence are often overlooked."
> > I think this inconsistency makes the paper difficult to create a reasonable and clear story for the motivation. It seems straightforward to describe the problem in the context of the time series forecasting task and compare it with existing techniques.

---

> > > ### Author Response · Authors · 2024-11-27
> > > **Following Up**
> > >
> > > Thank you for your review! We provide a summary of the changes made in our revision and shed some light on the reasons behind the modifications, for the reviewer’s convenience.
> > >
> > > We found it necessary to delete mentions of ASC as this is a very peripheral reference which we realise deviate from the paper's core focus. Our framework is a general one, hence we propose removing these sections for a clearer focus on our technical contributions, which will help readers better understand the scope and novelty of our work.
> > >
> > > We believe that the focus of the paper should be multi-task learning instead of forecasting due to the derivation of the loss function. To clarify, the foundation of VAE is rooted in reconstruction. However, our derived lower bound, derived from the joint log-likelihood, encompasses multiple tasks. The primary modification from the VAE architecture is forecasting. We would also like to clarify that the core part of our paper, specifically the derivation, remains unchanged.
> > >
> > > Additionally, forecasting results and literature have not been discarded, but merely moved to the appendix for more streamlined focus.
> > >
> > > We also found it necessary to add experiments on a new dataset to understand if a violation of one of our key assumptions is detrimental to the performance of the model. The new gait dataset has classes that are clearly correlated, which differs from our previous experiments on the electricity dataset where classes (hours, month, day-of-week) are independent of each other given the input. We thank reviewer kS2U for highlighting this possibility of an assumption not being met in a real-world dataset.
> > >
> > > The second figure we have added is regarding the fusion stage which was mentioned in the paper body previously, but not illustrated. We thank reviewer ETkb for pointing this out.
> > >
> > > We kindly request whether the reviewer requires further clarification or any additional explanation to support our approach.

---

### Official Review · Reviewer_kS2U · 2024-10-25

**Soundness:** 3
**Presentation:** 2
**Contribution:** 2
**Rating:** 6
**Confidence:** 4

**Summary:**

This paper proposes Deep-DIVE, a framework to learn disentangled features from the original input to form clusters in the embedding space and unify the classified features via the cross-attention mechanism. Experimental results on time series forecasting showcase that the proposed framework could disentangle the features and provide better forecasts than existing methods.

**Strengths:**

1. The paper is clearly written, easy to follow and understand.
2. Experiments compared with other baselines showcase that the proposed Deep-DIVE framework achieves better performance than existing baselines.
3. In terms of novelty, the DeepDIVE framework proposed in this work decomposes the latent space z into two distinct dimensions: marginal dimensions b and conditional dimensions a. The marginal dimensions b capture general trends and are independent of each other, while the conditional dimensions a are conditioned on b. This design enables b to capture shared patterns across conflicting tasks, while a learns task-specific features to avoid conflicts. Compared with existing methods, this approach better addresses the challenge of using a single variational encoding to model conflicting time series, resulting in improved performance and disentanglement.

**Weaknesses:**

1. In the introduction section, the author motivates the proposed Deep-DIVE framework by criticizing existing deep learning approaches for time series forecasting as being black-box in nature and hard to optimize. However, time series forecasting (TSF) is a well-established problem. The author should provide further explanation to better justify why the proposed framework is helpful in TSF.

2. The assumption 2 that $q_{\phi}(b_i,b_j|x) = q_{\phi}(b_i|x)q_{\phi}(b_j|x)$ for any i and j is too strong. Although I understand the author's remark that this simplifying assumption often works well in practice, some explanation or intuition about why it often works would be helpful.

**Questions:**

In the experiment results in section 5, Table 2, what does 'std' mean?

---

> ### Author Response · Authors · 2024-11-25
> **Official Comment by Authors**
>
> We thank the reviewer for their positive evaluation and encouraging feedback! We are pleased that they recognise the novelty and potential impact of our work on learning a disentangled representation for time-series forecasting.
>
> We have revised the manuscript to incorporate the insights gained from our discussion and add experiments. Below are our responses to weaknesses and questions through which we aim to address and alleviate any lingering concerns.
>
> **Addressing Weaknesses and Questions:**
> > In the introduction section, the author motivates the proposed Deep-DIVE framework by criticizing existing deep learning approaches for time series forecasting as being black-box in nature and hard to optimize. However, time series forecasting (TSF) is a well-established problem. The author should provide further explanation to better justify why the proposed framework is helpful in TSF.
>
> Thank you for pointing out this issue. We agree that the motivation behind our proposed framework still needs further clarification. As other reviewers have also raised similar queries, we have addressed this issue in the general reply to all reviewers. We have also placed it here for the reviewer’s convenience.
>
> We motivate the use of semi-supervised learning [in DeepDIVE] considering the fact that while it is common to employ a sequence as input to a time series forecasting model, other available labels that classify the entire sequence are often overlooked. For this task, the benefits of our variational encoding architecture are two-fold. Firstly, building upon the statistical foundations provided by the VAE for learning deep latent-variable models and corresponding inference models for reconstruction, DeepDIVE also provides a principled and well-founded semi-supervised framework for learning deep latent-variable models and corresponding inference models which is useful for reconstruction, forecasting, and classification. Secondly, we believe that probabilistic latent variable models such as VAEs hold great potential for facilitating autonomous planning and resource allocation in situations characterized by uncertainty, mirroring real-world scenarios.
>
> We have made the corresponding modifications to the introduction to reflect this focus.
>
>  > The assumption 2 that $q_\phi (b_i,b_j│x)=q_\phi (b_i│x) q_\phi (b_j│x)$ for any i and j is too strong. Although I understand the author's remark that this simplifying assumption often works well in practice, some explanation or intuition about why it often works would be helpful.
>
> Thank you for the insightful comment! Indeed, an intuitive explanation would greatly help readers better understand the relationship between this assumption and the main goal of developing a non-conflicting objective for the proposed framework. In the derivation, this naïve conditional independence assumption is used to split the total marginal Kullback-Leibler divergence $D_{KL}(q_\phi(\prod_{i=1}^{n_2} b_i|x) \parallel p_\theta(\prod_{i=1}^{n_2} b_i))$ into the sum of divergences for individual marginal dimensions $\sum_{i=1}^{n_2} D_{KL}(q_\phi(b_i|x) \parallel p_\theta(b_i))$, each of which shares a common minimizer with the cross entropy for that dimension. Thus, it would be more precise to say that the sum of divergences for individual marginal dimensions only approximates the total marginal Kullback-Leibler divergence, and this approximation is accurate when assumption 2 holds. We believe that in the case when assumption 2 does not hold, ie. some pairs are positively correlated while others are negatively correlated, the summation term may have an aggregating effect even if the estimates of the individual marginal divergences are inaccurate. We have added this intuition to the remarks after Assumption 2 in the revised manuscript.
>
> This review has also inspired us to question if we may be able to understand/observe the extent to which the model’s performance is affected from a violation of this assumption. We have newly added experimental results to contrast between 2 datasets, one of which has clearly correlated classes, while the other does not. We find that regardless of whether the naïve assumption holds (dataset: electricity) or not (dataset: gait), DeepDIVE performs better in terms of both forecast and reconstruction loss, as compared to the same architecture without the disentanglement.
>
> > In the experiment results in section 5, Table 2, what does 'std' mean?
>
> ‘std’ is short for standard deviation (of the RRSE). We will add its definition to Section 5 of the revised manuscript.

---

### Official Review · Reviewer_ubqL · 2024-11-01

**Soundness:** 2
**Presentation:** 1
**Contribution:** 2
**Rating:** 3
**Confidence:** 2

**Summary:**

The authors propose to extend the use of Variational Autoencoders to generative forecasting and develop a novel objective for training.

**Strengths:**

- The loss function seems to be novel and theoretically grounded.

**Weaknesses:**

- Presentation: The paper is not well-written, it is difficult to follow and understand the idea.
    - The abstract is not specific. Multiple concepts such as multi-task learning, disentanglement, Naive Bayes, and other technical details are presented without a clear, coherent relationship among them. I suggest focusing on a central contribution, such as developing a non-conflicting objective for multi-task learning, and clarifying how elements like disentanglement, Naive Bayes, cross-entropy, and RBF specifically contribute to this goal.
    - Line 46 of the introduction states model explainability to be the key contribution. However, the abstract does not discuss about explainability. I suggest emphasizing a central contribution and ensuring that all concepts throughout the paper align with it.
    -  Line 50 of the introduction mentions conflicting objectives, but a more detailed discussion of how multiple objectives in multi-task learning may conflict and providing examples of these conflicting objectives in the context of the paper would make this concept clearer.
- Baselines: No discussion or reference of the comparison baselines in Table 2 are AUTOCTS(-KDF/KDP), DsaNet, and MtGnn provided. Could the authors provide a brief discussion on these baselines and why they were considered for comparison? Also, why were more recent baselines such as DeepGLO, TCN, and TLAE (as discussed in section 2.1) not included in the comparison?

**Questions:**

- I couldn't find the paper referenced in section 2 (Wong et. al.). Is it published yet? Could the authors provide a link to the paper?

---

> ### Author Response · Authors · 2024-11-25
> **Official Comment by Authors**
>
> We are grateful that the reviewer acknowledges the theoretical foundations of our work.
>
> We have updated the manuscript to reflect our discussion and improve presentation. We hope these changes will contribute to the overall cohesiveness of the manuscript. Below are our responses to weaknesses and questions.
>
> **Addressing Weaknesses and Questions:**
> > The abstract is not specific. Multiple concepts such as multi-task learning, disentanglement, Naive Bayes, and other technical details are presented without a clear, coherent relationship among them. I suggest focusing on a central contribution, such as developing a non-conflicting objective for multi-task learning, and clarifying how elements like disentanglement, Naive Bayes, cross-entropy, and RBF specifically contribute to this goal.
>
> We agree with the reviewer’s observation that the derivations involve many assumptions and technical details, which readers may not find easy to follow.
>
> The elements specified by the reviewer mainly pertain to Propositions 3 to 6. By Proposition 3, the sum of divergences for individual marginal dimensions $\sum_{i=1}^{n_2} D_{KL}(q_\phi(b_i|x) \parallel p_\theta(b_i))$ approximates the total marginal Kullback-Leibler divergence $D_{KL}(q_\phi(\prod_{i=1}^{n_2} b_i|x) \parallel p_\theta(\prod_{i=1}^{n_2} b_i))$, and this approximation is accurate when the naïve conditional independence assumption holds (this was inspired by the Naïve Bayes algorithm). Further, Proposition 4 shows that the individual marginal dimensions $D_{KL}(q_\phi(b_i|x) \parallel p_\theta(b_i))$ are upper bounded by a term, where Propositions 5 and 6 together show that this term shares a common minimizer with the corresponding cross entropy for that dimension. However, this derivation requires that we choose a kernel that is log-concave, and the RBF kernel is proposed as an example of such a kernel. We feel that this set of derivations present a new perspective in guiding our splitting of the latent space into conditional and marginal dimensions, thus contributing to a disentangled latent space.
>
> We find that adding the above details to the abstract may be too specific, but we have newly added a graphical model in Section 3 to illustrate the role played by some of the key terms and assumptions in the derivations.
>
> > Line 46 of the introduction states model explainability to be the key contribution. However, the abstract does not discuss about explainability. I suggest emphasizing a central contribution and ensuring that all concepts throughout the paper align with it.
>
> We would like to clarify that our primary contribution lies in extending the original VAE architecture by incorporating labelled data for temporal forecasting tasks. Regarding the term “explainable”, we used it to emphasise that DeepDIVE provides mathematical foundations for understanding the model’s behaviour through rigorous theoretical analysis. To avoid ambiguity, we have removed the term “explainability” and instead focus on describing our theoretical contributions more precisely in the revised manuscript.
>
> > Line 50 of the introduction mentions conflicting objectives, but a more detailed discussion of how multiple objectives in multi-task learning may conflict and providing examples of these conflicting objectives in the context of the paper would make this concept clearer.
>
> Thank you for the helpful suggestion! We have newly added a brief discussion regarding conflicting objectives in multi-task learning problems in Section 1.
>
> > Baselines: No discussion or reference of the comparison baselines in Table 2 are AUTOCTS(-KDF/KDP), DsaNet, and MtGnn provided. Could the authors provide a brief discussion on these baselines and why they were considered for comparison? Also, why were more recent baselines such as DeepGLO, TCN, and TLAE (as discussed in section 2.1) not included in the comparison?
>
> We appreciate the reviewer bringing this benchmarking issue to our attention. We report that we have newly implemented TLAE with 4 GRU layers in the latent space and tested it on the electricity dataset. The results are presented in the Appendix. We demonstrate that using TCN along the temporal dimension to encode the input data allows DeepDIVE to learn a more representative posterior distribution as compared to TLAE, in which the encoding is performed along the features dimension to learn a global time series within the latent space, which further justifies the effectiveness of our theoretical framework.
>
> [continued in the next comment]

---

> > ### Author Response · Authors · 2024-11-25
> > **Official Comment by Authors**
> >
> > [continued from the previous comment]
> >
> > > I couldn't find the paper referenced in section 2 (Wong et. al.). Is it published yet? Could the authors provide a link to the paper?
> >
> > Thank you for highlighting that this paper currently seems to be unavailable online. At the present moment, we believe this paper has been accepted to the IEEE iSPEC conference but has not been published yet. However, in light of the reviews received, we have decided to focus the discussion on extending the original VAE architecture to include the use of labelled data for a forecasting use case. As such, we will remove this citation along with other peripheral references, for a clearer focus on our technical contributions, which will help readers better understand the scope and novelty of our work.

---

> > > ### Author Response · Authors · 2024-11-27
> > > **Following Up**
> > >
> > > Thank you for your review! With the deadline for manuscript revisions approaching, we are eager to confirm whether the changes we have made regarding the multi-objective problem, along with the added experiments and illustration in the appendix related to the derivation, have enhanced the overall cohesiveness of our manuscript and adequately addressed your concerns.

---

> > > > ### Comment · Reviewer_ubqL · 2024-12-02
> > > >
> > > > I would like to thank the authors for their response.
> > > >
> > > > > **"We find that adding the above details to the abstract may be too specific"**
> > > >
> > > > I didn't mean to add all the mathematical details. I meant to build coherence between in how the technical ideas are connected to the main goal of the paper.
> > > >
> > > > I appreciate the authors' effort in revising the paper within such a short timeframe with an additional baseline. Overall the writing looks better now. However, it can still be improved.
> > > >
> > > > -The introduction has improved. However, the discussion of previous works, their limitations, and the broader context remains inadequate. I recommend expanding the first paragraph to give more context.
> > > >
> > > >
> > > > - The first few sentences in the abstract lack coherence. While the opening sentence discusses the effective learning of a representative latent data space, the second jumps to describing the proposed disentanglement approach. To enhance cohesion, I suggest briefly explaining how disentanglement contributes to learning an effective latent space, as this appears to be the paper’s primary contribution.
> > > >
> > > > - >**Abstract: "We theoretically prove that combining the objectives for reconstruction and forecasting fully captures the lower bound and mathematically derive a loss function for disentanglement using Naive Bayes."**
> > > >     - The terms "Reconstruction and forecasting" are unclear as they have not been defined. It's better to write them as "combining multiple objectives".
> > > >
> > > > - No baselines from tables 1,2 and 3 are discussed in the "Related works" section.
> > > >
> > > > I agree with Reviewer N7k7 that, given the significant changes, the paper should undergo the full review process again. Accordingly, I will maintain my current score.

---

### Official Review · Reviewer_ETkb · 2024-11-01

**Soundness:** 2
**Presentation:** 2
**Contribution:** 2
**Rating:** 3
**Confidence:** 4

**Summary:**

This paper proposes a novel framework for enhancing the interpretability of representations learned from time-series prediction. The authors derive a variational evidence lower bound (ELBO) that extends the original VAE's ELBO to accommodate time-series forecasting scenarios, along with theoretical and implementation details for optimizing this learning objective. Specifically, the latent space is designed into separate marginal (denoted by "b") and conditional (denoted by "a") distributions that contribute to the calculation of the log-likelihood of both the look-back window and the forecasting window. Experiments are conducted on two time-series datasets, and both qualitative visualizations and quantitative metrics are evaluated for the proposed method.

**Strengths:**

(1) The detailed theoretical derivations of the proposed ELBO for time-series forecasting are solid and make sense, and it could be promising and inspiring for future research on learning disentangled representations in time-series prediction tasks.

(2) I appreciate the authors' efforts to report the variance (standard deviation) of the model's performance, which allows readers to better evaluate the results and comparisons.

**Weaknesses:**

The main weakness of this paper lies in its writing and evaluations.

(1) The writing lacks consistency between the text, figures, and equations. For example: (1a) While the inference and KL divergence for q(b∣x) is well-discussed, I had difficulties to find any description for inferring q(a∣x,b). In Fig. 1, it appears that a is directly computed from x, which contradicts the equation, and I couldn't find any other text to explain this. (1b) Although the "cross-attention mechanism" and "fusion stage" are mentioned in the abstract and introduction as part of this paper's contributions, they are never described or explained in the rest of this paper. It appears that the proposed framework is implemented just using an auto-encoder architecture, as suggested by Fig. 1 and line 412. Additionally, in line 518, the authors state, "Unlike attention maps and convolutional neural network (CNN) feature maps, DeepDIVE presents data representations in ...," which further increases the confusion about whether the proposed framework includes cross-attention. (1c) There are no descriptions or explanations in the main text for Fig. 3, and the figure caption is also limited.

(2) Although the combined effects of a and b in the latent space are evaluated in Fig. 2, separate evaluations and comparisons of a and b are missing, which I believe is critical for evaluating the proposed framework. Readers may be interested in understanding the differences between the representations learned in a and b, and how using separate a and b is beneficial compared to using just one latent variable. The authors are encouraged to conduct comprehensive experiments to demonstrate the representations learned in the marginal and conditional distributions in the latent space, separately, and how they impact the final prediction tasks. For example, the authors could consider using t-SNE plots for a and b individually and presenting traversal results for both of them to identify any expected differences.

**Questions:**

(1) Total correlation is also an option for promoting the independence of the latent variables. Besides using Naive Bayes, do the authors have any experience or insights in deriving and optimizing total correlation terms within the KL divergence? Additionally, how would this affect the model performance compared to using Naive Bayes? For instance, one approach for optimizing total correlation can be found in [1].

[1] Chen, Ricky TQ, et al. "Isolating sources of disentanglement in variational autoencoders." Advances in neural information processing systems 31 (2018).

---

> ### Author Response · Authors · 2024-11-25
> **Official Comment by Authors**
>
> We sincerely thank the reviewer for their careful reading of our manuscript and their positive assessment of our theoretical contributions. We are particularly encouraged that the reviewer sees the promise in our theoretical framework and its potential to inspire future research directions in disentangled representation learning for time-series forecasting.
>
> We have added experiments and literature following your suggestions. Below are our responses to weaknesses and questions which we hope can address and dispel any remaining concerns.
>
> **Addressing Weaknesses and Questions:**
> > While the inference and KL divergence for q(b∣x) is well-discussed, I had difficulties to find any description for inferring q(a∣x,b). In Fig. 1, it appears that a is directly computed from x, which contradicts the equation, and I couldn't find any other text to explain this.
>
> Thank you for pointing out this issue, we agree that Section 4 would benefit from a clearer connection between the frozen weights during main network backpropagation and the model learning dimensions a that are conditioned on b. We believe that this is achieved by freezing the RBF layers for the marginal dimensions, together with the weights in the last layer of the encoder that affect the marginal latent dimensions. The key here is to decouple the learning process of the marginal and conditional latent dimensions in an interleaving training scheme. Intuitively, this is similar to the alternating least squares algorithm [1], where the algorithm alternates between fixing the first factor when updating the second, and fixing the second factor when updating the first. However, one main difference is that in our case it is always the marginal dimensions that are fixed when the conditional dimensions are trained, so that it is always the conditional dimensions that are conditioned on the marginal ones. We have newly added this rationale as further elaboration on the last line of Section 4.
>
> [1] Dave Zachariah, Martin Sundin, Magnus Jansson, and Saikat Chatterjee. Alternating least-squares for low-rank matrix reconstruction. IEEE Signal Processing Letters, 19(4):231–234, 2012.
>
> > Although the "cross-attention mechanism" and "fusion stage" are mentioned in the abstract and introduction as part of this paper's contributions, they are never described or explained in the rest of this paper. It appears that the proposed framework is implemented just using an auto-encoder architecture, as suggested by Fig. 1 and line 412.
>
> Thank you for pointing out that the details of the model implementation have not been described, specifically the cross-attention mechanism in the fusion stage that was mentioned in the abstract. To elaborate, this fusion stage is the first layer of the decoder, directly after the RBF layer. In this stage, since the outputs of the RBF layer are of dimensions equal to the number of RBF units of that layer, we first standardize the dimensions with linear layers, before performing cross-attention with both the conditional dimension and RBF outputs as the query, and the conditional dimensions as the key and value. The remainder of the decoder after the fusion stage is only a simple multi-layer perceptron. However, we would like to clarify that similar to the original VAE, the architecture within the encoder and decoder are flexible. We have newly added this description along with an illustration of the fusion stage in Section 4.
>
> > Additionally, in line 518, the authors state, "Unlike attention maps and convolutional neural network (CNN) feature maps, DeepDIVE presents data representations in ...," which further increases the confusion about whether the proposed framework includes cross-attention.
>
> While we do use cross attention in the decoder portion of our network, we felt that for visualization purposes, the resulting attention maps from the attention mechanism are less useful than latent space embeddings.
>
> > There are no descriptions or explanations in the main text for Fig. 3, and the figure caption is also limited.
>
> Fig. 3 (density plot) is a continuation of the plot in Fig. 2 (scatter) illustrating the density of the latent embeddings along each marginal dimension. Compared to Fig. 2, in which there may be overlaps, Fig. 3 more clearly shows the distribution and concentration of data points along the marginal dimensions, for easier identification of class distributions along each dimension.
>
> [continued in the next comment]

---

> > ### Author Response · Authors · 2024-11-25
> > **Official Comment by Authors**
> >
> > [continued from the previous comment]
> >
> > > Although the combined effects of a and b in the latent space are evaluated in Fig. 2, separate evaluations and comparisons of a and b are missing, which I believe is critical for evaluating the proposed framework. Readers may be interested in understanding the differences between the representations learned in a and b, and how using separate a and b is beneficial compared to using just one latent variable. The authors are encouraged to conduct comprehensive experiments to demonstrate the representations learned in the marginal and conditional distributions in the latent space, separately, and how they impact the final prediction tasks. For example, the authors could consider using t-SNE plots for a and b individually and presenting traversal results for both of them to identify any expected differences.
> >
> > Thank you for the helpful suggestion! We also report that we have additionally implemented variants of DeepDIVE with only the conditional (a) or marginal (b) dimensions respectively, and added experimental results to Section 5.
> >
> > > Total correlation is also an option for promoting the independence of the latent variables. Besides using Naive Bayes, do the authors have any experience or insights in deriving and optimizing total correlation terms within the KL divergence? Additionally, how would this affect the model performance compared to using Naive Bayes? For instance, one approach for optimizing total correlation can be found in [1].
> >
> > We agree that this recommendation would strengthen our work, and help position it better in the existing research landscape.   We have implemented β-TCVAE with λ=0, β=6, and parameters (`include_mutinfo=True`, `tcvae=True`). For fairness of comparison, we use the same encoder and decoder that DeepDIVE uses for the implementation of β-TCVAE, and have also added logpy to the modified ELBO function that β-TCVAE uses, where the distributions of x, y and the prior are defined as $N(\bar{x},\sigma_x)$ and $N(\bar{y},\sigma_y)$ respectively. Similar to the original study in the cited paper, we find that this compromises the density estimation, as the reconstruction and forecast RRSE of the model trained on the modified ELBO function exhibits lower performance compared to DeepDIVE. We have newly added these results to Section 5.

---

> > > ### Comment · Reviewer_ETkb · 2024-11-25
> > >
> > > I appreciate the authors' response and look forward to a revised manuscript being uploaded to OpenReview. This revision should address all issues related to the figures and writing, as discussed above, before I can further evaluate the improvements in this manuscript.

---

> > > > ### Comment · Reviewer_ETkb · 2024-11-26
> > > >
> > > > I appreciate the authors' efforts in revising the paper. The presentation is improved compared to the first draft. However, it still does not meet the standard of ICLR. The idea of this paper is promising, and the authors are encouraged to further refine the writing, organization, and experiment settings for future submission. I will keep my score.

---

> > > > > ### Author Response · Authors · 2024-11-27
> > > > > **Following Up**
> > > > >
> > > > > Thank you for your review! We hope that the additional information and rationale in Section 4 helps to illustrate the use of cross-attention for merging the information from the conditional and marginal dimensions in the fusion stage, and clarify the link between the freezing of weights and conditioning of the conditional against the marginal dimensions. We also hope that the added experiments and literature review regarding $\beta$-TCVAE help to position our paper better in the literature.
> > > > >
> > > > > We welcome your suggestions for enhancing the manuscript and would appreciate specific feedback on potential improvements.

---

### Author Response · Authors · 2024-11-25
**Official Comment by Authors**

Dear Esteemed Reviewers,

We thank all reviewers for their thorough evaluation and valuable input to help improve the manuscript! We are gratified that most reviewers find our derivations novel (ETkb, ubqL, kS2U), theoretically grounded (ETkb, ubqL, N7k7) and inspiring (ETkb), although we also acknowledge that the presentation of our work needs improvement (ETkb, ubqL, N7k7). In light of the reviews received, to better help readers understand the impact of the work and its position with regards to other related works, we have decided to strive for more focused discussion regarding the extension of the original VAE architecture to include the use of labelled data for a forecasting use case. We motivate the use of semi-supervised learning considering the fact that while it is common to employ a sequence as input to a time series forecasting model, other available labels that classify the entire sequence are often overlooked. For this task, the benefits of our variational encoding architecture are two-fold. Building upon the statistical foundations provided by the VAE for learning deep latent-variable models and corresponding inference models for reconstruction, DeepDIVE also provides a principled and well-founded semi-supervised framework for learning deep latent-variable models and corresponding inference models which is useful for reconstruction, forecasting, and classification. We also believe that probabilistic latent variable models such as VAEs hold great potential for facilitating autonomous planning and resource allocation in situations characterized by uncertainty, mirroring real-world scenarios. We will address all remaining concerns and clarifications in an individual response. We thank the reviewers again for their time.

We are also currently modifying the manuscript and will publish the updated manuscript within approximately 15 hours.

---

> ### Author Response · Authors · 2024-11-26
> **Upload of Revised Manuscript**
>
> Dear Esteemed Reviewers,
>
> We have uploaded the revised manuscript to reflect our discussion and improve on the clarity and presentation. We would like to express our deepest gratitude to the reviewers for their time and patience.

---

### Meta-Review · Area_Chair_xEaq · 2024-12-17

**Metareview:**

This paper applies the KL term decomposition techniques from disentangled VAE literature to improve time-series forecasting. The idea is to disentangle the latent features, and in decoding part processing them individually before final combination via the cross-attention.

The major issues of this paper from reviewer point of view are (1) lack of ablation study and certain baselines, and (2) confusing presentation regarding the explanation of the design choices. Most reviewers agreed that the technical derivations are solid though.

While the overall techniques and the key equations are clear to understand, I think the paper would benefit from a rewrite and reordering the presentation. Personally I would present Section 4 first with more space to explain the latent variable structure, and then go back to section 3 for the training details. Also I personally recommend streamlining the theorem presentations and proofs (e.g., put them to appendix), and just focusing on explaining the core intuition of the key loss function. The proof technique for many of the other theorems/propositions -- manipulating the KL term -- is well known in disentangled VAE & variational inference literature.

PS: the revised version of this paper exceeds the page limit (10 pages) for the main text.

**Additional Comments On Reviewer Discussion:**

Author - Reviewer discussions: some clarification questions were addressed, but the majority of the reviewers were not convinced to increase their score.

AC - Reviewer discussions: I initiated the discussion but received no reply.

---

### Decision · Program_Chairs · 2025-01-22

Reject